# SHARPNESS-AWARE MINIMIZATION: GENERAL ANALYSIS AND IMPROVED RATES

**Dimitris Oikonomou**
CS & MINDS
Johns Hopkins University
doikono1@jh.edu

**Nicolas Loizou**
AMS & MINDS
Johns Hopkins University
nloizou1@jh.edu

## ABSTRACT

Sharpness-Aware Minimization (SAM) has emerged as a powerful method for improving generalization in machine learning models by minimizing the sharpness of the loss landscape. However, despite its success, several important questions regarding the convergence properties of SAM in non-convex settings are still open, including the benefits of using normalization in the update rule, the dependence of the analysis on the restrictive bounded variance assumption, and the convergence guarantees under different sampling strategies. To address these questions, in this paper, we provide a unified analysis of SAM and its unnormalized variant (USAM) under one single flexible update rule (Unified SAM), and we present convergence results of the new algorithm under a relaxed and more natural assumption on the stochastic noise. Our analysis provides convergence guarantees for SAM under different step size selections for non-convex problems and functions that satisfy the Polyak-Lojasiewicz (PL) condition (a non-convex generalization of strongly convex functions). The proposed theory holds under the arbitrary sampling paradigm, which includes importance sampling as special case, allowing us to analyze variants of SAM that were never explicitly considered in the literature. Experiments validate the theoretical findings and further demonstrate the practical effectiveness of Unified SAM in training deep neural networks for image classification tasks.

## 1  INTRODUCTION

Consider the classical finite-sum optimization problem

$$\min_{x \in \mathbb{R}^d} \left[ f(x) = \frac{1}{n} \sum_{i=1}^{n} f_i(x) \right], \tag{1}$$

where each $f_i$ is differentiable, $L_i$-smooth and lower bounded. Let $X^*$ be the set of minimizers of $f$, which we assume is non-empty. In practical scenarios, the variable $x$ represents the model parameters, $n$ is the total number of training instances, and the functions $f_i$ are loss functions that measure how close our model is to the $i$-th training data point. The goal is to minimize the average loss of all training instances.

Understanding the generalization capabilities of Deep Neural Networks (DNNs) is a central concern in machine learning research, (Zhang et al., 2016; Hardt et al., 2016; Neyshabur et al., 2017; Wilson et al., 2017; Neyshabur et al., 2018; Zhang et al., 2021). The training objective function $f$ has numerous stationary points that perfectly fit the training data (Liu et al., 2020), and each of these points can lead to dramatically varying generalization performances. This suggests that minimizing the training objective using a carefully selected optimization algorithm can lead to convergence at a point with improved generalization performance (Foret et al., 2021).

Recent studies have observed that the sharpness of the training loss, that is, how rapidly it changes in a neighborhood around the model parameters, correlates strongly with the generalization error (Keskar et al., 2016; Jiang et al., 2019; Singh et al., 2025; Dziugaite & Roy, 2018). This observation has motivated recent works (Foret et al., 2021; Zheng et al., 2021; Wu et al., 2020; Andriushchenko et al., 2023; Xie et al., 2024; Tahmasebi et al., 2024) aiming to minimize sharpness to improve

generalization. More specifically, building on these ideas Foret et al. (2021), proposed reformulating the optimization problem in (1) into a min-max problem of the following form:

$$\min_{x \in \mathbb{R}^d} \max_{\|\varepsilon\| \leq \rho} f(x + \varepsilon)$$

where $\varepsilon$ represents the radius of the desired neighborhood. The merits of such a formulation reside in the fact that essentially we minimize the empirical sharpness measure $\max_{\|\varepsilon\| \leq \rho} f(x + \varepsilon) - f(x)$, which inevitably will lead to flatter minima. The objective now is to find $x$ that minimizes $f$ not just at a specific point but across the entire $\varepsilon$-neighborhood. By taking the first-order Taylor expansion of $f$ around $x$ and solving for the optimal $\varepsilon^*$, the (Normalized) Sharpness-Aware Minimization (SAM) update rule is obtained:

$$x^{t+1} = x^t - \gamma_t \nabla f_{S_t} \left( x^t + \rho_t \frac{\nabla f_{S_t}(x^t)}{\|\nabla f_{S_t}(x^t)\|} \right), \tag{SAM}$$

where $S_t \subseteq [n]$ is a random subset of data points (mini-batch) with cardinality $\tau$ sampled independently at each iteration $t$. The normalization of the inner gradient ensures that the point $\tilde{x}^t = x^t + \rho_t \frac{\nabla f_{S_t}(x^t)}{\|\nabla f_{S_t}(x^t)\|}$ is a good approximation of $x^t$, since $\|\tilde{x}^t - x^t\| = \rho_t$. This leads to a more stable optimization process, as explained in Dai et al. (2023).

In an orthogonal direction and building upon SAM, Unnormalized Sharpness-Aware Minimization (USAM) was introduced by Andriushchenko & Flammarion (2022) and further investigated in Shin et al. (2024); Dai et al. (2023). The update rule for USAM is defined as follows:

$$x^{t+1} = x^t - \gamma_t \nabla f_{S_t} \left( x^t + \rho_t \nabla f_{S_t}(x^t) \right). \tag{USAM}$$

In contrast to SAM, USAM omits the normalization thus the point $\tilde{x}^t$ can be potentially far from $x^t$ making the $\tilde{x}^t$'s of USAM updates much larger. This means that the removal of normalization can lead to much more aggressive steps making the USAM potentially more unstable.

Although the two variants appear closely related, the proof techniques and upper bounds used in the convergence analysis of SAM are substantially different from those in USAM. Furthermore, the convergence guarantees of the two variants vary significantly. For example, in the deterministic setting (full-batch), SAM guarantees convergence only to a neighborhood of the solution, whereas USAM does not. Additionally, the step sizes $\gamma_t$ and $\rho_t$ used in the two update rules to guarantee convergence are very different. All of these differences motivate the importance and necessity of a novel general analysis of SAM-type algorithms, unifying the two main variants (SAM and USAM) and providing the ability to design and analyze new SAM-like methods filling existing gaps in the theoretical understanding of the update rules.

In this work we develop such unified framework that allows the combination of the two approaches and, at the same time, obtains the best-known convergence guarantees under relaxed assumptions.

**Main Contributions.** Our main contributions are summarized below.

◇ **Unified Framework.** We propose the Unified SAM, an update rule that is a convex combination of SAM and USAM, given by:

$$x^{t+1} = x^t - \gamma_t \nabla f_{S_t} \left( x^t + \rho_t \left( 1 - \lambda_t + \frac{\lambda_t}{\|\nabla f_{S_t}(x^t)\|} \right) \nabla f_{S_t}(x^t) \right) \tag{Unified SAM}$$

where $\lambda \in [0, 1]$. The new formulation captures both USAM and SAM as special cases ($\lambda = 0$ and $\lambda = 1$, respectively), but more importantly, it opens up a wide range of possible update rules beyond these traditional settings. The unified framework offers the flexibility to adjust the degree of normalization (using different values for $\lambda$) based on specific model needs, offering a more versatile approach to SAM.

◇ **Technical Assumptions on the Stochastic Noise.** Existing convergence analyses of stochastic SAM rely heavily on the bounded variance assumption, that is, there exists a $\sigma \geq 0$ such that $\mathbb{E} \|\nabla f_{S_t}(x) - \nabla f(x)\|^2 \leq \sigma^2$, (Andriushchenko & Flammarion, 2022; Si & Yun, 2023; Li & Giannakis, 2023; Harada & Iiduka, 2024; Mi et al., 2022; Zhuang et al., 2022) or sometimes to the much stronger bounded gradient condition $\mathbb{E} \|\nabla f_{S_t}(x)\|^2 \leq q^2$, where $q \geq 0$ (Mi et al., 2022; Zhuang et al., 2022). While these assumptions have been crucial in previous analyses, they can be

| Work | Assumptions | Arbitrary Sampling? | SAM Variant |
|------|-------------|---------------------|-------------|
| *PL functions* | | | |
| (Andriushchenko & Flammarion, 2022) | BV | ✗ | USAM |
| (Shin et al., 2024) | Interpolation | ✗ | USAM |
| (Dai et al., 2023) | Deterministic | ✗ | USAM |
| Theorems 3.2 and 3.5 | ER | ✓ | Unified SAM |
| *General Non-convex functions* | | | |
| (Mi et al., 2022) | BV, BG | ✗ | SAM/SSAM |
| (Zhuang et al., 2022) | BV, BG | ✗ | GSAM |
| (Andriushchenko & Flammarion, 2022) | BV | ✗ | USAM |
| (Li & Giannakis, 2023) | BV | ✗ | SAM |
| (Si & Yun, 2023) | BV | ✗ | SAM |
| Theorem 3.7 | ER | ✓ | Unified SAM |

Table 1: Summary of the convergence results in the SAM literature. In all works, smoothness is assumed. The top part of the table is for PL functions and the lower part is for general non-convex functions. Here BV = Bounded Variance, BG = Bounded Gradients.

overly restrictive. In the literature of convergence analysis for stochastic gradient descent (SGD), there have been a lot of efforts recently on relaxing such assumptions (Gower et al., 2019; Khaled & Richtárik, 2020; Gower et al., 2021), but, to date, no work has successfully used similar ideas for the analysis of SAM. In our analysis, we relax the bounded gradients/variance assumptions by utilizing the recently proposed Expected Residual (ER) condition Gower et al. (2021); Khaled & Richtárik (2020). As we explain later, in several scenarios, including smooth non-convex problems, ER holds for free and allows us to provide step sizes for SAM related to the sampling strategies.

◇ **Convergence guarantees for Unified SAM.** We provide tight convergence guarantees for Unified SAM, for smooth functions satisfying the Polyak-Lojasiewicz (PL) condition (Polyak, 1987; Lojasiewicz, 1963; Karimi et al., 2016) and for general non-convex functions. See also Table 1 for a summary of our results and comparison with closely related works.

- *PL functions:* For constant step-sizes $\gamma$ and $\rho$ we prove linear convergence for Unified SAM to a neighborhood of the solution. Our theorem holds without requiring the much stronger assumptions of interpolation condition or the bounded variance assumption of previous works (Andriushchenko & Flammarion, 2022; Shin et al., 2024). Additionally, we prove that for decreasing step sizes $\gamma_t$ and $\rho_t$, the Unified SAM converges to the exact solution with a sublinear $O(1/t)$ rate (under the ER condition). Our theoretical results hold under the arbitrary sampling paradigm and, as such, can capture tight convergence guarantees in the deterministic setting. For PL functions in the deterministic setting (full-batch SAM), we show that USAM converges to the exact solution, while SAM does not. This observation was first noted by Si & Yun (2023) for deterministic algorithms. To the best of our knowledge, our work is the first that provides tight convergence guarantees, showing this behaviour as a special case of stochastic algorithms.

- *Non-convex functions:* Under the ER condition, we show that for general non-convex functions Unified SAM with step sizes that depend on $T$ (the total number of iterations) achieves $\mathbb{E}\|\nabla f(x^T)\| < \varepsilon$ for a given $\varepsilon$ at a sublinear rate. This is the first result that drops the bounded variance assumption for both USAM and SAM, (Andriushchenko & Flammarion, 2022; Li & Giannakis, 2023), and substitutes it with the Expected Residual condition.

Finally, as corollaries of the main theorems for the above two classes of problems, we obtain the state-of-the-art convergence guarantees for SGD (a special case of SAM with $\rho = 0$), showing the tightness of our analysis.

◇ **Arbitrary Sampling.** Via a stochastic reformulation of the finite sum problem (1), firstly introduced in Gower et al. (2019), we explain how our convergence guarantees of Unified SAM hold under the arbitrary sampling paradigm. This allows us to cover a wide range of samplings for USAM and SAM (and their convex combination via $\lambda \in [0, 1]$) that were never considered in the literature before, including uniform sampling and importance sampling as special cases. In this sense, our analysis of Unified SAM is unified for different sampling strategies.

$\diamond$ **Numerical Evaluation.** In Section 4, we present extensive experiments validating different aspects of our theoretical results (behavior of methods in the deterministic setting, importance sampling, and different step-size selections). We also assess the performance of Unified SAM in training deep neural networks for multi-class image classification problems. An open-source implementation of our method is available at https://github.com/dimitris-oik/unifiedsam.

## 2    UNIFIED SAM WITH ARBITRARY SAMPLING

In this work, we provide a theoretical analysis of Unified SAM that allows us to obtain convergence guarantees of any minibatch and reasonable sampling selection.

We are able to do that by leveraging the recently proposed "stochastic reformulation" of the minimization problem (1) from Gower et al. (2019; 2021). Following an identical setting to Gower et al. (2021), we assume that we have access to unbiased gradient estimates $g(x) \in \mathbb{R}^d$ such that $\mathbb{E}[g(x)] = \nabla f(x)$. For example, we can have $g(x) = \frac{1}{\tau} \sum_{i \in S} \nabla f_i(x)$ to be a mini-batch, where $S \subseteq [n]$ is chosen uniformly at random with $|S| = \tau$. To accommodate any form of mini-batching, we utilize the arbitrary sampling notation $g(x) = \nabla f_v(x) := \frac{1}{n} \sum_{i=1}^n v_i \nabla f_i(x)$, where $v \in \mathbb{R}^n$ is a random sampling vector drawn from a distribution $\mathcal{D}$ such that $\mathbb{E}_{\mathcal{D}}[v_i] = 1$, for $i = 1, \ldots, n$. Then the original problem (1) can be reformulated as $\min_{x \in \mathbb{R}^d} \mathbb{E}_{\mathcal{D}} \left[ f_v(x) := \frac{1}{n} \sum_{i=1}^n v_i f_i(x) \right]$. Note that it follows immediately from the definition of sampling vector that $\mathbb{E}[g(x)] = \frac{1}{n} \sum_{i=1}^n \mathbb{E}[v_i] \nabla f_i(x) = \nabla f(x)$. Using this reformulation of the original problem, the update rule of Unified SAM can be rewritten as follows:

$$x^{t+1} = x^t - \gamma_t g \left( x^t + \rho_t \left( 1 - \lambda_t + \frac{\lambda_t}{\|g(x^t)\|} \right) g(x^t) \right), \qquad \text{(Unified SAM)}$$

where $g(x^t) \sim \mathcal{D}$ is sampled i.i.d at each iteration and $\rho_t \geq 0$, $\gamma_t > 0$ and $\lambda_t \in [0, 1]$. The name unified stems from the fact that the update rule indeed unifies both USAM and SAM, however, we acknowledge that there exist other SAM-like variants that our approach does not cover.

**Arbirtary Sampling.** Using the stochastic reformulation, the update rule of Unified SAM includes several variants of the algorithm, each related to different sampling, by simply varying the distribution $\mathcal{D}$ (that satisfies $\mathbb{E}_{\mathcal{D}}[v_i] = 1, \forall i \in [n]$). This flexibility implies that different choices of $\mathcal{D}$ lead to distinct SAM-type methods (never proposed in the literature before) for solving the original problem (1). In this work we focus on two representative sampling distributions, without aiming to be exhaustive:

1. **Single element sampling:** We choose only singleton sets $S = \{i\}$ for $i \in [n]$, i.e. $\mathbb{P}[|S| = 1] = 1$. Each number $i$ is sampled with probability $p_i \in [0, 1]$ or more formally the vector $v \in \mathbb{R}^n$ is defined via $\mathbb{P}[v = e_i/p_i] = p_i$, where $\sum_{i=1}^n p_i = 1$. It is clear that $\mathbb{E}[v_i] = 1$. For example, when $p_i = 1/n$ for all $i$ then this reduces to the well-known *uniform sampling*.

2. **$\tau$-nice Sampling:** Let $\tau \in [n]$. We generate a random subset $S \subseteq [n]$ by choosing uniformly from all subsets of size $\tau$. More formally, the vector $v \in \mathbb{R}^n$ is defined by $\mathbb{P}[v = \frac{n}{\tau} \sum_{i \in S} e_i] := 1/\binom{n}{\tau} = \frac{\tau!(n-\tau)!}{n!}$ for any subset $S \subseteq [n]$ with $|S| = \tau$. Using a double counting argument, one can show that $\mathbb{E}[v_i] = 1$ (see Gower et al. (2019)).

Importantly, our analysis applies to all forms of mini-batching and supports various choices of sampling vectors $v$. Later in Section 3.4, we provide additional details on non-uniform single element sampling strategies. In addition, it is clear that if $\tau = n$ in the $\tau$-nice sampling then we recover the full batch or deterministic regime. Later in Section 3.2, we further demonstrate how our analysis encompasses deterministic SAM and USAM as special cases.

## 3    CONVERGENCE ANALYSIS

In this section, we present our main convergence results. Firstly, we introduce the main assumption for our results, namely the Expected Residual (ER) condition. Then we focus on PL functions, where we demonstrate a linear convergence rate using constant step sizes, and also provide a variant with decreasing step sizes for convergence to the exact solution. Moreover, we extend the analysis to general non-convex functions. Lastly, we discuss the use of importance sampling.

## 3.1 MAIN ASSUMPTION

In all our theoretical results, we rely on the Expected Residual condition.

**Assumption 3.1** (Expected Residual Condition). Let $f^{\mathrm{inf}} = \inf_{x \in \mathbb{R}^n} f(x)$. We say the Expected Residual condition holds if there exist parameters $A, B, C \geq 0$ such that for an unbiased estimator $g(x)$ of $\nabla f(x)$, we have that for all $x \in \mathbb{R}^d$

$$\mathbb{E}_{\mathcal{D}} \|g(x)\|^2 \leq 2A[f(x) - f^{\mathrm{inf}}] + B\|\nabla f(x)\|^2 + C. \tag{ER}$$

Most prior works in the SAM literature assume either bounded gradients (e.g., Mi et al. (2022); Zhuang et al. (2022)) or bounded variance (e.g., Andriushchenko & Flammarion (2022); Harada & Iiduka (2024)). Both conditions are stronger assumptions than ER. Note that the bounded gradients assumption is captured by ER for $A = B = 0$, $C > 0$, while the bounded variance is obtained for $A = 0$, $B = 1$, and $C > 0$. For a detailed analysis of other conditions that automatically satisfy ER, see Gower et al. (2021) and Khaled & Richtárik (2020). Finally, when each $f_i$ is $L_i$-smooth under mild assumptions on the distribution $\mathcal{D}$, one can show that ER holds immediately (not an assumption but property of the problem) and has closed-from expressions for the constants A, B, and C. For more details on the expressions A, B, and C in this scenario, please check Appendix B.

## 3.2 PL FUNCTIONS

One of the popular generalizations of strong convexity in the literature is the Polyak-Lojasiewicz (PL) condition, (Karimi et al., 2016; Lei et al., 2019). We call a function $\mu$-PL if $\|\nabla f(x)\|^2 \geq 2\mu(f(x) - f(x^*))$, for all $x \in \mathbb{R}^d$. In the following, we prove linear convergence of Unified SAM for $\mu$-PL functions.

**Theorem 3.2.** *Assume that each $f_i$ is $L_i$-smooth, $f$ is $\mu$-PL and the ER is satisfied. Set $L_{\max} = \max_{i \in [n]} L_i$. Then the iterates of Unified SAM with*

$$\rho_t = \rho \leq \rho^* := \frac{\mu}{L_{\max}(\mu + 2[B\mu + A](1-\lambda)^2)}, \gamma_t = \gamma \leq \gamma^* := \frac{\mu - L_{\max}\rho(\mu + 2[B\mu + A](1-\lambda)^2)}{2L_{\max}(B\mu + A)[2L_{\max}^2\rho^2(1-\lambda)^2 + 1]},$$

*satisfy:*
$$\mathbb{E}[f(x^t) - f(x^*)] \leq (1 - \gamma\mu)^t \left[f(x^0) - f(x^*)\right] + N,$$

*where $N = \frac{L_{\max}}{\mu} \left(C\gamma + \rho(1 + 2\gamma L_{\max}^2\rho)\right) \left[\lambda^2 + C(1-\lambda)^2\right]$.*

As an immediate corollary of Theorem 3.2, we get the following guarantees for USAM and SAM, when we substitute $\lambda = 0$ and $\lambda = 1$ respectively.

**Corollary 3.3.** *Consider the same assumptions as in Theorem 3.2.*

- **USAM:** *For USAM with $\rho \leq \frac{\mu}{L_{\max}(\mu + 2[B\mu + A])}$ and $\gamma \leq \frac{\mu - L_{\max}\rho(\mu + 2[B\mu + A])}{2L_{\max}(B\mu + A)[2L_{\max}^2\rho^2 + 1]}$, it holds:*

$$\mathbb{E}[f(x^t) - f(x^*)] \leq (1 - \gamma\mu)^t \left[f(x^0) - f(x^*)\right] + \frac{L_{\max}C}{\mu} \left(\gamma + \rho(1 + 2\gamma L_{\max}^2\rho)\right).$$

- **SAM:** *For SAM with $\rho_t = \rho \leq \frac{1}{L_{\max}}$ and $\gamma_t = \gamma \leq \frac{\mu(1 - L_{\max}\rho)}{2L_{\max}(B\mu + A)}$, it holds:*

$$\mathbb{E}[f(x^t) - f(x^*)] \leq (1 - \gamma\mu)^t \left[f(x^0) - f(x^*)\right] + \frac{L_{\max}}{\mu} \left(C\gamma + \rho(1 + 2\gamma L_{\max}^2\rho)\right).$$

To the best of our knowledge, all prior convergence results for (stochastic) SAM have relied on the strong assumption of bounded variance, as seen in works like Andriushchenko & Flammarion (2022), Si & Yun (2023), Li & Giannakis (2023) and Harada & Iiduka (2024). In contrast, the above theorem is the first to establish convergence for SAM without relying on this assumption. The closest related works on the convergence of constant step size SAM for PL functions are Shin et al. (2024) and Dai et al. (2023). The former provides a linear convergence rate for USAM in the *interpolated* regime, while the latter establishes a linear convergence rate for USAM in the *deterministic* regime.

Our result is the first to demonstrate linear convergence in the fully stochastic regime. Additionally, when $\rho = 0$, Unified SAM reduces to SGD, and Theorem 3.2 recovers the step sizes and rates (up to constants) of Theorem 4.6 from Gower et al. (2021), demonstrating the tightness of our results.

Recall that with our general analysis via the arbitrary sampling framework, the proposed convergence guarantees hold for the $\tau$-nice sampling as well (see Section 2). As such, a special case of Theorem 3.2 is the case where $|S| = n$ with probability one. That is, we run Unified SAM with a full batch, $\tau = n$. In this scenario, using Proposition B.3, the ER condition holds with $A = 0$, $B = 1$ and $C = 0$ and Theorem 3.2 simplifies as follows.

**Corollary 3.4** (Deterministic SAM). *Assume that each $f_i$ is $L_i$-smooth and $f$ is $\mu$-PL. Then the iterates of the deterministic Unified SAM with $\rho \leq \frac{1}{L_{\max}(1+2(1-\lambda)^2)}$ and $\gamma \leq \frac{1-L_{\max}\rho\left(1+2(1-\lambda)^2\right)}{2L_{\max}[2L_{\max}^2\rho^2(1-\lambda)^2+1]}$ satisfy:*

$$\mathbb{E}[f(x^t) - f(x^*)] \leq (1-\gamma\mu)^t \left[f(x^0) - f(x^*)\right] + \frac{L_{\max}\rho(1 + 2\gamma L_{\max}^2\rho)\lambda^2}{\mu}.$$

First, observe that the PL parameter $\mu$ no longer appears in the step sizes $\rho$ and $\gamma$. Additionally, when $\lambda = 0$, i.e. USAM, the method converges to the exact solution at a linear rate. However, for $\lambda > 0$, and in particular when $\lambda = 1$ (SAM), convergence is only up to a neighborhood. This suggests that even in the deterministic setting, SAM does not fully converge to the minimum. This was first investigated in Si & Yun (2023) and a similar result appear in Dai et al. (2023). We illustrate this phenomenon experimentally in Section 4.1.

Finally, as an extension of Theorem 3.2, we also show how to obtain convergence to exact solution with an $O(1/t)$ rate for Unified SAM using decreasing step sizes.

**Theorem 3.5.** *Assume that each $f_i$ is $L_i$-smooth, $f$ is $\mu$-PL and the ER is satisfied. Then the iterates of Unified SAM with $\rho_t = \min\left\{\frac{1}{2t+1}, \rho^*\right\}$ and $\gamma_t = \min\left\{\frac{2t+1}{(t+1)^2\mu}, \gamma^*\right\}$, where $\rho^*$ and $\gamma^*$ are defined in Theorem 3.2, satisfy: $E[f(x^t) - f(x^*)] \leq O\left(\frac{1}{t}\right)$.*

The detailed expression hidden under the big $O$ notation can be found in Appendix C. A similar result appears in Andriushchenko & Flammarion (2022) where they provide decreasing step size selection for USAM for PL functions and prove a convergence rate of $O(1/t)$. However, their result relies on the additional assumption of bounded variance. In contrast, our theorem does not require this assumption and is valid for any $\lambda \in [0, 1]$. Notably, to the best of our knowledge, this is the first decreasing step size result for SAM.

### 3.3 GENERAL NON-CONVEX FUNCTIONS

In this section, we remove the PL assumption and work with general non-convex functions. First, we start with a general proposition that upper bounds the quantity $\min_{t=0,...,T-1} \mathbb{E}\|\nabla f(x^t)\|^2$, which can serve as an intermediate result for Theorem 3.7. Our approach follows a similar derivation to the analysis of SGD in the same setting by Khaled & Richtárik (2020).

**Proposition 3.6.** *Assume that each $f_i$ is $L_i$-smooth and the ER is satisfied. Then the iterates of Unified SAM with $\rho \leq \min\left\{\frac{1}{4L_{\max}}, \frac{1}{8BL_{\max}(1-\lambda)^2}\right\}$ and $\gamma \leq \frac{1}{8BL_{\max}}$, satisfy:*

$$\min_{t=0,...,T-1} \mathbb{E}\|\nabla f(x^t)\|^2 \leq \frac{2\left(1 + 2A\gamma L_{\max}\left[\rho(1-\lambda)^2(1+2\gamma\rho L_{\max}^2) + \gamma\right]\right)^T}{T\gamma}[f(x^0) - f^{\inf}]$$
$$+ 2L_{\max}\left[C\gamma + \rho(1 + 2\gamma\rho L_{\max}^2)(\lambda^2 + C(1-\lambda)^2)\right].$$

In order to control the coefficient of the term $f(x^0) - f^{\inf}$ of Proposition 3.6 from exploding we need to carefully select the step sizes $\rho$ and $\gamma$. This is what the following Theorem 3.7 achieves.

**Theorem 3.7.** *Let $\varepsilon > 0$ and set $\delta_0 = f(x_0) - f^{\mathrm{inf}} \geq 0$. For $\rho = \overline{\rho}$ and $\gamma = \overline{\gamma}$,[a] provided that*

$$T \geq \frac{\delta_0 L_{\max}}{\varepsilon^2} \max \left\{ 96B, 24(1-\lambda)\sqrt{3L_{\max}A}, \frac{5184 L_{\max} A^2 (1-\lambda)^4 \delta_0}{\varepsilon^2}, \frac{864 \delta_0 A}{\varepsilon^2}, \frac{144C}{\varepsilon^2}, \right.$$
$$\left. \frac{288 L_{\max}^2 (1-\lambda)^2}{\varepsilon^2} \right\}$$

*the iterates of Unified SAM satisfy $\min_{t=0,\ldots,T-1} \mathbb{E} \|\nabla f(x^t)\| \leq \varepsilon$.*

---

[a]For the precise expressions of $\overline{\rho}$ and $\overline{\gamma}$ we refer to Theorem C.4.

The results in Proposition 3.6 and Theorem 3.7 are tight, as setting $\rho = 0$ Unified SAM reduces in SGD and these simplify to the step sizes and rates (up to constants) of Theorem 2 and Corollary 1 from Khaled & Richtárik (2020). Other results for general non-convex functions can be found in Mi et al. (2022), Zhuang et al. (2022) and Li & Giannakis (2023) for SAM and in Andriushchenko & Flammarion (2022) for USAM. However, as mentioned earlier, all these analyses rely on the strong assumption of bounded variance and/or bounded gradients. In contrast, our result uses weaker assumptions and offers guarantees for any $\lambda \in [0, 1]$. Furthermore, Khanh et al. (2024) have results for general non-convex functions in the deterministic setting though all their results are asymptotic. Another closely related work is Nam et al. (2023), where they also assume the expected residual condition, however, they additionally assume bounded gradient of $f$ and their results are only asymptotic and hold almost surely.

### 3.4 Beyond Uniform Sampling: Importance Sampling

In this work, we provide theorems in the PL and non-convex regime through which we can analyze all importance sampling and mini-batch variants of Unified SAM (and as a result of USAM and SAM). We achieve that via our general arbitrary sampling analysis. In this section, we focus on the non-convex regime (Theorem 3.7) and provide importance sampling for Unified SAM with single-element sampling. That is, we select probabilities that optimize the iteration complexity (lower bound of $T$). In particular, to derive the importance sampling probabilities, we substitute the bounds for $A$, $B$, and $C$ from Proposition B.2 into Theorem 3.7, resulting in:

$$T \geq \frac{\delta_0 L_{\max}}{\varepsilon^2} \max \left\{ 24(1-\lambda)\sqrt{\frac{3L_{\max}}{n\tau} \max_i \frac{L_i}{p_i}}, \frac{\frac{5184 L_{\max}}{n^2\tau^2}(\max_i \frac{L_i}{p_i})^2 (1-\lambda)^4 \delta_0}{\varepsilon^2}, \right.$$
$$\left. \frac{\frac{864 \delta_0}{n\tau} \max_i \frac{L_i}{p_i}}{\varepsilon^2}, \frac{\frac{288}{n\tau} \max_i \frac{L_i}{p_i} \sigma^*}{\varepsilon^2}, \frac{288 L_{\max}^2 (1-\lambda)^2}{\varepsilon^2} \right\}$$

Having substituted the values of $A$, $B$, and $C$ in the above expression, to obtain the best bound, we require optimizing the quantity $\max_i \frac{L_i}{np_i}$ over the probabilities $p_i$. This results in $p_i = \frac{L_i}{\sum_{j=1}^n L_j}$. Similar probabilities have been proposed for several optimization algorithms, including SGD (Gower et al., 2019; 2021; Khaled & Richtárik, 2020), variance-reduced methods (Khaled et al., 2023), and methods for min-max optimization (Gorbunov et al., 2022; Loizou et al., 2020; Choudhury et al., 2024; Beznosikov et al., 2023). To the best of our knowledge, our work is the first to provide importance sampling for SAM algorithms.

## 4 Numerical Experiments

In this section, we evaluate our proposed step sizes for Unified SAM on both deterministic and stochastic PL problems, with experiments designed to illustrate our theoretical findings. Additionally, we explore different values of $\lambda$ when training deep neural networks to improve accuracy.

### 4.1 Validation of the Theory

In this part, we empirically validate our theoretical results and illustrate the main properties of Unified SAM that our theory suggests in Section 3. In these experiments, we focus on $\ell_2$-regularized regression problems (problems with strongly convex objective $f$ and components $f_i$ and thus PL)

and we evaluate the performance of Unified SAM on synthetic data. The loss function of the $\ell_2$-regularized *ridge regression* is given by

$$f(x) = \frac{1}{2}\|\boldsymbol{A}x - b\|^2 + \lambda_r \|x\|^2 = \frac{1}{2n}\sum_{i=1}^{n}\left(\boldsymbol{A}[i,:]x - b_i\right)^2 + \frac{\lambda_r}{2}\|x\|^2$$

and the loss function of the $\ell_2$-regularized *logistic regression* is given by

$$f(x) = \frac{1}{2n}\sum_{i=1}^{n}\log\left(1 + \exp\left(-b_i\boldsymbol{A}[i,:]x\right)\right) + \frac{\lambda_r}{2}\|x\|^2.$$

In both problems $\boldsymbol{A} \in \mathbb{R}^{n \times d}$, $b \in \mathbb{R}^n$ are the given data and $\lambda_r \geq 0$ is the regularization parameter.

**Normalized SAM converges only in neighborhood.** In Corollary 3.4 we highlight that our theoretical results indicate that in the deterministic case, USAM achieves linear convergence to the exact solution. In contrast, when $\lambda > 0$, and specifically for SAM ($\lambda = 1$), convergence is only to a neighborhood of the solution, an unusual outcome for deterministic optimization methods. We validate this observation experimentally in Figure 1. We run a ridge regression problem with $n = 100, d = 100, \lambda_r = 0$. The matrix $\boldsymbol{A}$ has been generated according to Lenard & Minkoff (1984) such that the condition number of $A$ is 10 and the vector $b$ has been sampled from the standard Gaussian distribution. We have used the deterministic Unified SAM for $\lambda = 0.0, \ldots, 1.0$ and we run each algorithm for 50 epochs. Indeed we can see that USAM ($\lambda = 0$) converges all the way to the exact solution while the other choices of $\lambda$ converge to a neighborhood. It is also noteworthy that the neighborhood increases as $\lambda$ approaches 1.

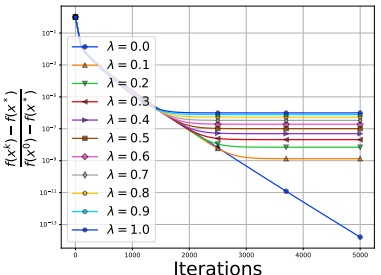

Figure 1: Deterministic Unified SAM for various values of $\lambda$ applied to the ridge regression problem. USAM ($\lambda = 0$) converges to the exact solution while the other variants $\lambda > 0$ converge to a neighborhood of the solution.

**Constant vs Decreasing Step size.** In this part, we compare the performance of Unified SAM under both constant and decreasing step size regimes, as described in Theorem 3.2 and Theorem 3.5, respectively. For this experiment, we consider a logistic regression task with $n = 100$, $d = 100$, and $\lambda_r = 3/n$. As before the matrix $\boldsymbol{A}$ has been generated such that its condition number is 10 and the vector $b$ has been sampled from the standard Gaussian distribution. We run Unified SAM with $\lambda = 0.0, 0.5, 1.0$, using uniform single-element sampling for $10,000$ epochs across 5 trials. The average results of these trials, along with one standard deviation, are shown in Figure 2. Initially, the trajectory of the decreasing step size follows that of the constant step size. However, as the constant step size version of Theorem 3.2 approaches a neighborhood near optimality, it stagnates. In contrast, the decreasing step size from Theorem 3.5 allows for continued improvement, leading to better overall accuracy.

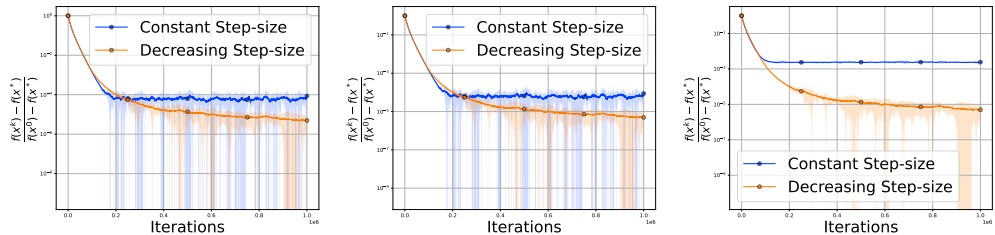

Figure 2: Comparison between constant and decreasing step size regimes of Unified SAM. From left to right we have $\lambda = 0.0, 0.5, 1.0$

**Uniform vs Importance Sampling.** In this experiment, we demonstrate the benefit of using importance sampling compared to uniform sampling. We consider a ridge regression problem with $n = 100$, $d = 100$, and $\lambda_r = 3/n$. The eigenvalues of matrix $\boldsymbol{A}$ are sampled uniformly from the interval $[1.0, 10.0]$, and the vector $b$ is drawn from a standard Gaussian distribution. We run Unified SAM with $\lambda = 0.0, 0.5, 1.0$, employing single-element uniform sampling for $3,000$ epochs across

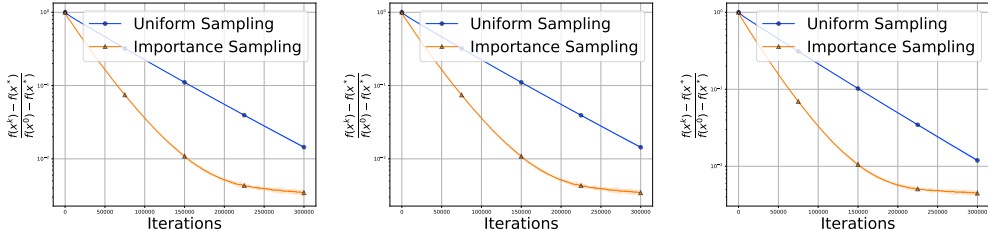

Figure 3: Comparison between uniform and importance sampling for Unified SAM. From left to right we have $\lambda = 0.0, 0.5, 1.0$

five trials. For importance sampling, the probabilities are set as $p_i = L_i / \sum_{j=1}^n L_j$. The averaged results with one standard deviation are presented in Figure 3. The results clearly show that importance sampling enhances the convergence of Unified SAM for all values of $\lambda$, which is consistent with the discussion in Section 3.4.

## 4.2 IMAGE CLASSIFICATION

In this section, we evaluate the generalization performance of Unified SAM using the classic image classification task. The models are trained on the CIFAR-10 and CIFAR-100 datasets, (Krizhevsky et al., 2009). For data augmentation, we apply standard techniques such as random crop, random horizontal flip, and normalization (DeVries, 2017). All experiments are conducted on NVIDIA RTX 6000 Ada GPUs.

**Models.** We use several models in our experiments, including ResNet-18 (RN-18) (He et al., 2016a), and PreActResNet-18 (PRN-18) (He et al., 2016b). To demonstrate the scalability of Unified SAM, we also include Wider ResNet (WRN-28-10) (Zagoruyko & Komodakis, 2016).

**Different values of $\lambda$.** This subsection explores how varying $\lambda$ in the Unified SAM update rule affects generalization performance. We experiment with the PRN-18 and WRN-28-10 models trained on CIFAR-10 and CIFAR-100. The Unified SAM method is trained using $\rho = 0.1, 0.2, 0.3, 0.4$ and $\lambda_t = 0.0, 0.5, 1.0, 1/t, 1 - 1/t$. The last two choices of $\lambda_t$ can be intuitively explained as follows: for $\lambda_t = 1 - 1/t$, the algorithm starts as USAM ($\lambda_1 = 0$) and gradually transitions toward SAM as $\lambda_t \to 1$ when $t \to \infty$, meaning it begins with USAM and approaches SAM. Conversely, for $\lambda_t = 1/t$, the algorithm behaves the other way around, starting closer from SAM and converging to USAM over time. Following Pang et al. (2021) and Zhang et al. (2024), we set the weight decay to $5 \times 10^{-4}$, the momentum to 0.9, and train for 100 epochs. The step size $\gamma$ is initialized at 0.1 and reduced by a factor of 10 at the 75-th and 90-th epochs.

Each experiment is repeated three times, and we report the averages and standard errors in Tables 2 and 3 and Tables 8 and 9 in appendix. The results indicate that, for a fixed $\rho$, the optimal $\lambda$ value is not always $\lambda = 0$ or $\lambda = 1$. Overall, $\lambda_t = 1 - 1/t$ appears to be a reliable choice. In particular, we observe that USAM ($\lambda = 0$) does not have the best performance in any of these experiments. On the contrary in the CIFAR10 dataset the value $\lambda_t = 1 - 1/t$ is a good choice while in the CIFAR100 dataset both $\lambda_t = 1$ and $\lambda_t = 1 - 1/t$ offer strong performance.

Table 2: Test accuracy (%) of Unified SAM for WRN-28-10 on CIFAR10, evaluated across different values of $\rho$ and $\lambda$. With bold we highlight the best performance for fixed $\rho$.

| **Unified SAM** | $\lambda = 0.0$ | $\lambda = 0.5$ | $\lambda = 1.0$ | $\lambda = 1/t$ | $\lambda = 1 - 1/t$ |
|---|---|---|---|---|---|
| $\rho = 0.1$ | $95.7_{\pm 0.01}$ | $95.68_{\pm 0.11}$ | $\mathbf{95.9}_{\pm 0.08}$ | $95.84_{\pm 0.07}$ | $95.81_{\pm 0.03}$ |
| $\rho = 0.2$ | $95.8_{\pm 0.05}$ | $95.77_{\pm 0.09}$ | $95.93_{\pm 0.07}$ | $95.71_{\pm 0.13}$ | $\mathbf{95.98}_{\pm 0.1}$ |
| $\rho = 0.3$ | $95.35_{\pm 0.3}$ | $95.88_{\pm 0.1}$ | $95.95_{\pm 0.09}$ | $95.68_{\pm 0.02}$ | $\mathbf{95.99}_{\pm 0.06}$ |
| $\rho = 0.4$ | $95.46_{\pm 0.02}$ | $95.76_{\pm 0.1}$ | $95.62_{\pm 0.05}$ | $95.46_{\pm 0.27}$ | $\mathbf{95.79}_{\pm 0.07}$ |
| **SGD** | | | $95.35_{\pm 0.06}$ | | |

**Unified VaSSO.** The Variance-Suppressed Sharpness-Aware Optimization (VaSSO) method, introduced in Li & Giannakis (2023), is an extension of SAM designed to reduce variance in gradient estimates. VaSSO adjusts the direction of gradient updates using a combination of past gradients and the current gradient, controlled by a parameter $\theta$, aiming to suppress noise during training and enhance

Table 3: Test accuracy (%) of Unified SAM for WRN-28-10 on CIFAR100, evaluated across different values of $\rho$ and $\lambda$. With bold we highlight the best performance for fixed $\rho$.

| Unified SAM | $\lambda = 0.0$ | $\lambda = 0.5$ | $\lambda = 1.0$ | $\lambda = 1/t$ | $\lambda = 1 - 1/t$ |
|---|---|---|---|---|---|
| $\rho = 0.1$ | $80.84_{\pm 0.08}$ | $\mathbf{81.01}_{\pm 0.11}$ | $80.69_{\pm 0.11}$ | $80.81_{\pm 0.24}$ | $80.88_{\pm 0.31}$ |
| $\rho = 0.2$ | $81.12_{\pm 0.18}$ | $81.45_{\pm 0.23}$ | $\mathbf{81.53}_{\pm 0.09}$ | $81.31_{\pm 0.21}$ | $81.22_{\pm 0.19}$ |
| $\rho = 0.3$ | $80.94_{\pm 0.13}$ | $81.64_{\pm 0.16}$ | $81.62_{\pm 0.16}$ | $81.03_{\pm 0.14}$ | $\mathbf{81.71}_{\pm 0.17}$ |
| $\rho = 0.4$ | $80.1_{\pm 0.22}$ | $81.31_{\pm 0.16}$ | $\mathbf{81.70}_{\pm 0.06}$ | $80.39_{\pm 0.07}$ | $81.59_{\pm 0.05}$ |
| **SGD** | | | $79.79_{\pm 0.18}$ | | |

generalization, particularly in overparameterized models. In their work, they prove that VaSSO converges at the same asymptotic rate as SAM, under the bounded variance assumption. The update rule of VaSSO is defined as follows: $d_t = (1 - \theta)d_{t-1} + \theta \nabla f_i(x^t)$, $x^{t+1} = x^t - \gamma_t \nabla f_i \left( x^t + \rho_t \frac{d_t}{\|d_t\|} \right)$. We can incorporate our unification approach, initially designed for SAM, into VaSSO. This leads to the following modified update rule:

$$d_t = (1 - \theta)d_{t-1} + \theta \nabla f_i(x^t)$$

$$x^{t+1} = x^t - \gamma_t \nabla f_i \left( x^t + \rho_t \left( 1 - \lambda_t + \frac{\lambda_t}{\|d_t\|} \right) d_t \right) \qquad \text{(Unified VaSSO)}$$

Note that when $\lambda_t = 1$ then we recover VaSSO as introduced in Li & Giannakis (2023).

In this section, we conduct experiments to evaluate the performance of Unified VaSSO. All models are trained for 200 epochs with a batch size of 128. A cosine scheduler is employed in all cases, with an initial step size of $0.05$. The weight decay is set to $0.001$. For VaSSO, we use $\theta = 0.4$, as this value provides the best accuracy according to Li & Giannakis (2023). For the CIFAR-10 dataset, we set $\rho = 0.1$, while for CIFAR-100, we use $\rho = 0.2$. Each experiment is repeated three times, and we report the average of the maximum test accuracy along with the standard error. The numerical results are presented in Tables 4 and 5.

The results show that, for the WideResNet-28-10 model the value $\lambda_t = 1 - 1/t$ produces the best accuracy. For the ResNet-18 model the best values for $\lambda$ appear to be $\lambda = 0.5$ and $\lambda = 1 - 1/t$. Thus, the choice $\lambda_t = 1 - 1/t$ appears to be a strong overall choice. In any cases, a $\lambda$ value different from 1 achieves the best performance.

Table 4: Test accuracy (%) of Unified VaSSO on various neural networks trained on CIFAR10.

| Model | $\lambda = 0.0$ | $\lambda = 0.5$ | $\lambda = 1.0$ | $\lambda = 1/t$ | $\lambda = 1 - 1/t$ |
|---|---|---|---|---|---|
| ResNet-18 | $96.12_{\pm 0.02}$ | $96.10_{\pm 0.05}$ | $96.22_{\pm 0.11}$ | $96.03_{\pm 0.04}$ | $\mathbf{96.34}_{\pm 0.01}$ |
| WideResNet-28-10 | $96.77_{\pm 0.07}$ | $96.93_{\pm 0.03}$ | $97.03_{\pm 0.06}$ | $96.71_{\pm 0.06}$ | $\mathbf{97.06}_{\pm 0.09}$ |

Table 5: Test accuracy (%) of Unified VaSSO on various neural networks trained on CIFAR100.

| Model | $\lambda = 0.0$ | $\lambda = 0.5$ | $\lambda = 1.0$ | $\lambda = 1/t$ | $\lambda = 1 - 1/t$ |
|---|---|---|---|---|---|
| ResNet-18 | $79.82_{\pm 0.15}$ | $\mathbf{80.01}_{\pm 0.07}$ | $79.76_{\pm 0.03}$ | $79.93_{\pm 0.13}$ | $79.86_{\pm 0.04}$ |
| WideResNet-28-10 | $83.07_{\pm 0.25}$ | $83.29_{\pm 0.33}$ | $83.51_{\pm 0.09}$ | $82.83_{\pm 0.18}$ | $\mathbf{83.66}_{\pm 0.19}$ |

## 5 CONCLUSION

In this work, we introduced Unified SAM, an algorithm that generalizes SAM and USAM via the convex combination parameter $\lambda \in [0, 1]$. The proposed analysis relaxes the common bounded variance assumption used in previous works by employing the ER condition, which enables us to prove convergence under weaker assumptions. In particular, under the ER, we established convergence guarantees for Unified SAM for both PL and general non-convex functions, and our analysis encompasses importance sampling and $tau$-nice sampling. Future work could investigate the optimal value of $\lambda$ for various classes of problems, extend the unified approach to incorporate adaptive step sizes $\rho_t$ and $\gamma_t$, and study Unified SAM in distributed and federated learning settings. Another promising direction is to explore how our approach may help minimize the sharpness-aware loss by employing the Hessian-regularized methods proposed by Wen et al. (2023) and Bartlett et al. (2023).

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

# Supplementary Material

The Supplementary Material is organized as follows: In Appendix A, we give the basic definitions and lemmas that we need for the proofs. Appendix C presents the proofs of the theoretical guarantees from the main paper. In Appendix D, we provide additional experiments.

CONTENTS

## A TECHNICAL PRELIMINARIES

### A.1 BASIC DEFINITIONS

In this section, we present some basic definitions we use throughout the paper.

**Definition A.1** ($L$-smooth). A differentiable function $f : \mathbb{R}^d \to \mathbb{R}$ is $L$-smooth if there exists a constant $L > 0$ such that

$$\|\nabla f(x) - \nabla f(y)\| \leq L\|x - y\|, \tag{2}$$

or equivalently

$$f(x) - f(y) \leq \langle \nabla f(y), x - y \rangle + \frac{L}{2}\|x - y\|^2$$

for all $x, y \in \mathbb{R}^d$.

**Lemma A.2.** *Let $f$ be an $L$-smooth function. Then for all $x \in \mathbb{R}^n$ we have*

$$\|\nabla f(x)\|^2 \leq 2L[f(x) - f^{\inf}].$$

**Definition A.3** ($\mu$-PL). We say that a differentiable function $f : \mathbb{R}^d \to \mathbb{R}$ satisfies the Polyak-Lojasiewicz condition if

$$\|\nabla f(x)\|^2 \geq 2\mu(f(x) - f(x^*)), \tag{3}$$

for all $x \in \mathbb{R}^d$.

**Definition A.4** (Interpolation). We say that the interpolation condition holds if there exists $x^* \in \mathcal{X}^*$ such that

$$\min_{x \in \mathbb{R}^n} f_i(x) = f_i(x^*)$$

for all $i = 1, \ldots, n$.

**Proposition A.5** (Young's Inequality). *For any $a, b \in \mathbb{R}^n$ and $\beta \neq 0$ we have*

$$|\langle a, b \rangle| \leq \frac{1}{2\beta}\|a\|^2 + \frac{\beta}{2}\|b\|^2. \tag{4}$$

*Completing the square we have*

$$\|a + b\|^2 \leq (1 + \beta^{-1})\|a\|^2 + (1 + \beta)\|b\|^2. \tag{5}$$

*For $\beta = 1$ we get*

$$\|a + b\|^2 \leq 2\|a\|^2 + 2\|b\|^2. \tag{6}$$

### A.2 BASIC LEMMAS

**Lemma A.6.** *Assume that each $f_i$ is $L_i$-smooth and consider the iterates of Unified SAM. Then it holds*

$$\mathbb{E}_{\mathcal{D}} \left\| g\left(x^t + \rho_t\left(1 - \lambda_t + \frac{\lambda_t}{\|g(x^t)\|}\right)g(x^t)\right) \right\|^2 \leq 4L_{\max}^2\rho_t^2\lambda_t^2$$

$$+ 2\left[2L_{\max}^2\rho_t^2(1 - \lambda_t)^2 + 1\right]\mathbb{E}_{\mathcal{D}}\|g(x^t)\|^2.$$

*Proof.* We have

$$\mathbb{E}_{\mathcal{D}} \left\| g\left(x^t + \rho_t \left(1 - \lambda_t + \frac{\lambda_t}{\|g(x^t)\|}\right) g(x^t)\right) \right\|^2$$

$$= \mathbb{E}_{\mathcal{D}} \left\| g\left(x^t + \rho_t \left(1 - \lambda_t + \frac{\lambda_t}{\|g(x^t)\|}\right) g(x^t)\right) - g(x^t) + g(x^t) \right\|^2$$

$$\stackrel{(6)}{\leq} 2\,\mathbb{E}_{\mathcal{D}} \left\| g\left(x^t + \rho_t \left(1 - \lambda_t + \frac{\lambda_t}{\|g(x^t)\|}\right) g(x^t)\right) - g(x^t) \right\|^2 + 2\,\mathbb{E}_{\mathcal{D}} \|g(x^t)\|^2$$

$$\stackrel{(2)}{\leq} 2L_{\max}^2 \rho_t^2\,\mathbb{E}_{\mathcal{D}} \left\| (1-\lambda_t) g(x^t) + \frac{\lambda_t}{\|g(x^t)\|} g(x^t) \right\|^2 + 2\,\mathbb{E}_{\mathcal{D}} \|g(x^t)\|^2$$

$$\stackrel{(6)}{\leq} 4L_{\max}^2 \rho_t^2 (1-\lambda_t)^2\,\mathbb{E}_{\mathcal{D}} \|g(x^t)\|^2 + 4L_{\max}^2 \rho_t^2 \lambda_t^2 + 2\,\mathbb{E}_{\mathcal{D}} \|g(x^t)\|^2$$

$$= 4L_{\max}^2 \rho_t^2 \lambda_t^2 + 2\left[2L_{\max}^2 \rho_t^2 (1-\lambda_t)^2 + 1\right] \mathbb{E}_{\mathcal{D}} \|g(x^t)\|^2.$$

This completes the proof. $\qquad\square$

**Lemma A.7.** *Assume that each $f_i$ is $L_i$-smooth and consider the iterates of [Unified SAM](#). Then for any $\beta > 0$ it holds*

$$\mathbb{E}_{\mathcal{D}} \left\langle g\left(x^t + \rho_t \left(1 - \lambda_t + \frac{\lambda_t}{\|g(x^t)\|}\right) g(x^t)\right), \nabla f(x^t) \right\rangle \geq \left(1 - \frac{\beta}{2}\right) \|\nabla f(x^t)\|^2 - \frac{L_{\max}^2 \rho_t^2 \lambda_t^2}{\beta}$$
$$- \frac{L_{\max}^2 \rho_t^2 (1-\lambda_t)^2}{\beta}\,\mathbb{E}_{\mathcal{D}} \|g(x^t)\|^2.$$

*In particular, for $\beta = L_{\max}\rho_t$ we get*

$$\mathbb{E}_{\mathcal{D}} \left\langle g\left(x^t + \rho_t \left(1 - \lambda_t + \frac{\lambda_t}{\|g(x^t)\|}\right) g(x^t)\right), \nabla f(x^t) \right\rangle \geq \left(1 - \frac{L_{\max}\rho_t}{2}\right) \|\nabla f(x^t)\|^2$$
$$- L_{\max}\rho_t \lambda_t^2$$
$$- L_{\max}\rho_t (1-\lambda_t)^2\,\mathbb{E}_{\mathcal{D}} \|g(x^t)\|^2.$$

*Proof.* We have

$$\mathbb{E}_{\mathcal{D}} \left\langle g\left(x^t + \rho_t \left(1 - \lambda_t + \frac{\lambda_t}{\|g(x^t)\|}\right) g(x^t)\right), \nabla f(x^t) \right\rangle$$

$$= \mathbb{E}_{\mathcal{D}} \left\langle g\left(x^t + \rho_t \left(1 - \lambda_t + \frac{\lambda_t}{\|g(x^t)\|}\right) g(x^t)\right) - g(x^t), \nabla f(x^t) \right\rangle$$
$$+ \mathbb{E}_{\mathcal{D}} \langle g(x^t), \nabla f(x^t) \rangle$$

$$= \mathbb{E}_{\mathcal{D}} \left\langle g\left(x^t + \rho_t \left(1 - \lambda_t + \frac{\lambda_t}{\|g(x^t)\|}\right) g(x^t)\right) - g(x^t), \nabla f(x^t) \right\rangle + \|\nabla f(x^t)\|^2.$$

Now

$$-\mathbb{E}_{\mathcal{D}} \left\langle g\left(x^t + \rho_t \left(1 - \lambda_t + \frac{\lambda_t}{\|g(x^t)\|}\right) g(x^t)\right) - g(x^t), \nabla f(x^t) \right\rangle$$

$$\stackrel{(4)}{\leq} \frac{1}{2\beta}\,\mathbb{E}_{\mathcal{D}} \left\| g\left(x^t + \rho_t \left(1 - \lambda_t + \frac{\lambda_t}{\|g(x^t)\|}\right) g(x^t)\right) - g(x^t) \right\|^2$$
$$+ \frac{\beta}{2}\,\mathbb{E}_{\mathcal{D}} \|\nabla f(x^t)\|^2$$

$$\stackrel{(2)}{\leq} \frac{L_{\max}^2 \rho_t^2}{2\beta}\,\mathbb{E}_{\mathcal{D}} \left\| (1-\lambda_t) g(x^t) + \frac{\lambda_t}{\|g(x^t)\|} g(x^t) \right\|^2 + \frac{\beta}{2} \|\nabla f(x^t)\|^2$$

$$\stackrel{(6)}{\leq} \frac{L_{\max}^2 \rho_t^2}{2\beta} 2(1-\lambda_t)^2\,\mathbb{E}_{\mathcal{D}} \|g(x^t)\|^2 + \frac{L_{\max}^2 \rho_t^2}{2\beta} 2\lambda_t^2 + \frac{\beta}{2} \|\nabla f(x^t)\|^2$$

$$= \frac{L_{\max}^2 \rho_t^2 \lambda_t^2}{\beta} + \frac{L_{\max}^2 \rho_t^2 (1 - \lambda_t)^2}{\beta} \mathbb{E}_{\mathcal{D}} \|g(x^t)\|^2 + \frac{\beta}{2} \|\nabla f(x^t)\|^2,$$

thus

$$\mathbb{E}_{\mathcal{D}} \left\langle g\left(x^t + \rho_t \left(1 - \lambda_t + \frac{\lambda_t}{\|g(x^t)\|}\right) g(x^t)\right), \nabla f(x^t) \right\rangle \geq -\frac{L_{\max}^2 \rho_t^2 (1 - \lambda_t)^2}{\beta} \mathbb{E}_{\mathcal{D}} \|g(x^t)\|^2$$
$$- \frac{L_{\max}^2 \rho_t^2 \lambda_t^2}{\beta} - \frac{\beta}{2} \|\nabla f(x^t)\|^2$$
$$+ \|\nabla f(x^t)\|^2$$
$$= \left(1 - \frac{\beta}{2}\right) \|\nabla f(x^t)\|^2 - \frac{L_{\max}^2 \rho_t^2 \lambda_t^2}{\beta}$$
$$- \frac{L_{\max}^2 \rho_t^2 (1 - \lambda_t)^2}{\beta} \mathbb{E}_{\mathcal{D}} \|g(x^t)\|^2.$$

This completes the proof. $\qquad\square$

**Lemma A.8.** *Let $(r_t)_{t \geq 0}$ and $(\delta_t)_{t \geq 0}$ be sequences of non-negative real numbers and let $g > 1$ and $N \geq 0$. Assume that the following recursive relationship holds:*

$$r_t \leq g\delta_t - \delta_{t+1} + N \tag{7}$$

*Then it holds*

$$\min_{0 \leq t \leq T-1} r_t \leq \frac{g^T}{T} \delta_0 + N.$$

*Proof.* Set $g_t = g^{-t}$ for any $t \in \mathbb{Z}$. Then multiply both sides of (7) with $g_t$. This yields

$$g_t r_t \leq g_{t-1}\delta_t - g_t\delta_{t+1} + Ng_t.$$

Summing for $t = 0, \dots, T$ and telescoping we get

$$\sum_{t=0}^{T-1} g_t r_t \leq g_{-1}\delta_0 - g_{T-1}\delta_T + N \sum_{t=0}^{T-1} g_t$$
$$\leq g_{-1}\delta_0 + N \sum_{t=0}^{T-1} g_t. \tag{8}$$

Now let $G = \sum_{t=0}^{T-1} g_t$. Note that the sequence $(g_t)$ is decreasing for $t \geq 0$, thus $G \geq Tg_{T-1}$. Now dividing (8) by $G$ we get

$$\min_{0 \leq t \leq T-1} r_t \leq \frac{1}{G} \sum_{t=0}^{T-1} g_t r_t$$
$$\leq \frac{g_{-1}\delta_0}{G} + N$$
$$\leq \frac{g_{-1}\delta_0}{Tg_{T-1}} + N$$
$$= \frac{g^T \delta_0}{T} + N.$$

This completes the proof. $\qquad\square$

# B  MORE ON THE EXPECTED RESIDUAL CONDITION

In this section, we introduce several assumptions under which ER is automatically satisfied. Similar derivations are provided in Khaled & Richtárik (2020), where the authors focus on the analysis of SGD. Specifically, we consider the following cases:

- Assuming **bounded gradients**, i.e.

$$\mathbb{E}_{\mathcal{D}} \|g(x)\|^2 \leq \sigma^2, \quad \forall x \in \mathbb{R}^n,$$

  then the ER holds with $A = 0, B = 0, C = \sigma^2$.

- Assuming **bounded variance**, i.e.

$$\mathbb{E}_{\mathcal{D}} \|g(x) - \nabla f(x)\| \leq \sigma^2, \quad \forall x \in \mathbb{R}^n,$$

  then the ER holds with $A = 0, B = 1, C = \sigma^2$.

- Assuming **expected smoothness**, i.e.

$$\mathbb{E}_{\mathcal{D}} \|g(x) - \nabla f(x)\| \leq 2\mathcal{L}[f(x) - f^{\inf}], \quad \forall x \in \mathbb{R}^n,$$

  then the ER holds with $A = 2\mathcal{L}, B = 0, C = 0$.

- Assuming the **relaxed strong growth condition**, i.e.

$$\mathbb{E}_{\mathcal{D}} \|g(x)\| \leq \rho\|\nabla f(x)\| + \sigma^2, \quad \forall x \in \mathbb{R}^n,$$

  then the ER holds with $A = 0, B = \rho, C = \sigma^2$.

- Assuming the **relaxed strong growth condition**, i.e.

$$\mathbb{E}_{\mathcal{D}} \|g(x)\| \leq \alpha[f(x) - f^{\inf}] + \sigma^2, \quad \forall x \in \mathbb{R}^n,$$

  then the ER holds with $A = \alpha, B = 0, C = \sigma^2$.

Additionally, if we have more information about the problem and the distribution $\mathcal{D}$, we can derive stronger bounds on $A$, $B$, and $C$. The more assumptions we make, the tighter these bounds become, as demonstrated in the following propositions.

**Proposition B.1** (Prop. 2, (Khaled & Richtárik, 2020)). *Assume that each $f_i$ is $L_i$-smooth and that $\mathbb{E}_{\mathcal{D}}[v_i^2] < \infty$ for all $i \in [n]$. Let $\sigma^* = \frac{1}{n}\sum_{i=1}^n f_i(x^*) - f_i^* \geq 0$, where $f_i^* = \inf_{x \in \mathbb{R}^n} f_i(x)$. Then ER holds with $A = \max_i L_i \mathbb{E}_{\mathcal{D}}[v_i^2]$, $B = 0$ and $C = 2A\sigma^*$.*

This proposition indicates that if each $f_i$ is $L_i$-smooth and minimal assumptions hold for the distribution $\mathcal{D}$, then ER is satisfied. As an immediate corollary of Proposition B.1, if the problem further satisfies the interpolation assumption (see Definition A.4), then $C = 0$. The next proposition gives tighter constants for $A$, $B$, and $C$ in the context of the sampling strategies considered in this paper.

**Proposition B.2** (Prop. 3, (Khaled & Richtárik, 2020)). *Assume that each $f_i$ is $L_i$-smooth.*

1. *For the single element sampling ER holds with $A = \frac{1}{n\tau} \max_i \frac{L_i}{p_i}$, $B = 0$ and $C = 2A\sigma^*$.*

2. *For the $\tau$-nice sampling ER holds with $A = \frac{n-\tau}{\tau(n-1)} L_{\max}$, $B = \frac{n(\tau-1)}{\tau(n-1)}$ and $C = 2A\sigma^*$, where $L_{\max} = \max_i L_i$.*

Lastly, if we assume $x^*$-convexity with $\tau$-nice sampling, we obtain the following constants:

**Proposition B.3** (Prop. 3.3, (Gower et al., 2021)). *Assume that each $f_i$ is $L_i$-smooth and that there exists $x^* \in \mathcal{X}^*$ such that $f_i$ is $x^*$–convex. In addition, assume that $\mathcal{D}$ is the $\tau$-nice sampling. Then ER holds with $A = \frac{n-\tau}{\tau(n-1)} L_{\max}$, $B = 1$ and $C = \frac{2(n-\tau)}{\tau(n-1)}\sigma_1$, where $\sigma_1 = \sup_{x^* \in \mathcal{X}^*} \frac{1}{n}\sum_{i=1}^n \|\nabla f_i(x^*)\|^2$.*

## C PROOFS OF THE MAIN RESULTS

In this section we present the proofs of the main theoretical results presented in the main paper, i.e., the convergence guarantees of Unified SAM for PL and smooth functions and general non-convex and smooth functions. We restate the main theorems here for completeness.

In all cases, we use the convention $1/0 = \infty$.

### C.1 PROOF OF THEOREM 3.2

**Theorem C.1.** *Assume that each $f_i$ is $L_i$-smooth, $f$ is $\mu$-PL and the ER is satisfied. Set $L_{\max} = \max_{i \in [n]} L_i$. Then the iterates of Unified SAM with*

$$\rho_t = \rho \le \rho^* := \frac{\mu}{L_{\max}(\mu + 2[B\mu + A](1-\lambda)^2)}, \gamma_t = \gamma \le \gamma^* := \frac{\mu - L_{\max}\rho\left(\mu + 2[B\mu + A](1-\lambda)^2\right)}{2L_{\max}(B\mu + A)[2L_{\max}^2\rho^2(1-\lambda)^2 + 1]},$$

*satisfy:*

$$\mathbb{E}[f(x^t) - f(x^*)] \le (1 - \gamma\mu)^t \left[f(x^0) - f(x^*)\right] + N,$$

*where $N = \frac{L_{\max}}{\mu}\left(C\gamma + \rho(1 + 2\gamma L_{\max}^2\rho)\left[\lambda^2 + C(1-\lambda)^2\right]\right)$.*

*Proof of Theorem 3.2.* By combining the smoothness of function f with the update rule of Unified SAM we obtain:

$$f(x^{t+1}) \le f(x^t) + \langle \nabla f(x^t), x^{t+1} - x^t \rangle + \frac{L_{\max}}{2}\|x^{t+1} - x^t\|^2$$

$$= f(x^t) - \gamma_t \left\langle \nabla f(x^t), g\left(x^t + \rho_t\left(1 - \lambda_t + \frac{\lambda_t}{\|g(x^t)\|}\right)g(x^t)\right)\right\rangle$$

$$+ \frac{L_{\max}\gamma_t^2}{2}\left\|g\left(x^t + \rho_t\left(1 - \lambda_t + \frac{\lambda_t}{\|g(x^t)\|}\right)g(x^t)\right)\right\|^2.$$

By taking expectation conditioned on $x^t$ we obtain:

$$\mathbb{E}[f(x^{t+1}) - f(x^*)|x^t] - [f(x^t) - f(x^*)]$$

$$\le -\gamma_t \mathbb{E}_{\mathcal{D}}\left\langle g\left(x^t + \rho_t\left(1 - \lambda_t + \frac{\lambda_t}{\|g(x^t)\|}\right)g(x^t)\right), \nabla f(x^t)\right\rangle$$

$$+ \frac{L_{\max}\gamma_t^2}{2}\mathbb{E}_{\mathcal{D}}\left\|g\left(x^t + \rho_t\left(1 - \lambda_t + \frac{\lambda_t}{\|g(x^t)\|}\right)g(x^t)\right)\right\|^2$$

$$\overset{\text{Lem. A.6 and A.7}}{\le} -\gamma_t\left[\left(1 - \frac{L_{\max}\rho_t}{2}\right)\|\nabla f(x^t)\|^2 - L_{\max}\rho_t\lambda_t^2 - L_{\max}\rho_t(1-\lambda_t)^2\mathbb{E}_{\mathcal{D}}\|g(x^t)\|^2\right]$$

$$+ \frac{\gamma_t^2 L_{\max}}{2}\left[4L_{\max}^2\rho_t^2\lambda_t^2 + 2\left[2L_{\max}^2\rho_t^2(1-\lambda_t)^2 + 1\right]\mathbb{E}_{\mathcal{D}}\|g(x^t)\|^2\right]$$

$$= -\gamma_t\left(1 - \frac{L_{\max}\rho_t}{2}\right)\|\nabla f(x^t)\|^2$$

$$+ \gamma_t L_{\max}\left[\rho_t(1-\lambda_t)^2 + 2\gamma_t L_{\max}^2\rho_t^2(1-\lambda_t)^2 + \gamma_t\right]\mathbb{E}_{\mathcal{D}}\|g(x^t)\|^2$$

$$+ \gamma_t L_{\max}\rho_t\lambda_t^2(1 + 2\gamma_t L_{\max}^2\rho_t)$$

$$\overset{ER}{\le} -\gamma_t\left(1 - \frac{L_{\max}\rho_t}{2}\right)\|\nabla f(x^t)\|^2$$

$$+ 2A\gamma_t L_{\max}\left[\rho_t(1-\lambda_t)^2 + 2\gamma_t L_{\max}^2\rho_t^2(1-\lambda_t)^2 + \gamma_t\right]\left[f(x^t) - f(x^*)\right]$$

$$+ B\gamma_t L_{\max}\left[\rho_t(1-\lambda_t)^2 + 2\gamma_t L_{\max}^2\rho_t^2(1-\lambda_t)^2 + \gamma_t\right]\|\nabla f(x^t)\|^2$$

$$+ C\gamma_t L_{\max}\left[\rho_t(1-\lambda_t)^2 + 2\gamma_t L_{\max}^2\rho_t^2(1-\lambda_t)^2 + \gamma_t\right] + \gamma_t L_{\max}\rho_t\lambda_t^2(1 + 2\gamma_t L_{\max}^2\rho_t)$$

$$= -\gamma_t\left(1 - \frac{L_{\max}\rho_t}{2} - BL_{\max}\left[\rho_t(1-\lambda_t)^2 + 2\gamma_t L_{\max}^2\rho_t^2(1-\lambda_t)^2 + \gamma_t\right]\right)\|\nabla f(x^t)\|^2$$

$$+ 2A\gamma_t L_{\max} \left[ \rho_t(1-\lambda_t)^2 + 2\gamma_t L_{\max}^2 \rho_t^2(1-\lambda_t)^2 + \gamma_t \right] \left[ f(x^t) - f(x^*) \right]$$
$$+ \gamma_t L_{\max} \left[ C\rho_t(1-\lambda_t)^2 + 2C\gamma_t L_{\max}^2 \rho_t^2(1-\lambda_t)^2 + C\gamma_t + \rho_t \lambda_t^2 + 2\gamma_t L_{\max}^2 \rho_t^2 \lambda_t^2 \right].$$
$$(9)$$

In order to use the fact that is $\mu$-PL we need to ensure that the coefficient of $\|\nabla f(x^t)\|^2$ is non-negative. For this we have

$$1 - \frac{L_{\max}\rho_t}{2} - BL_{\max} \left[ \rho_t(1-\lambda_t)^2 + 2\gamma_t L_{\max}^2 \rho_t^2(1-\lambda_t)^2 + \gamma_t \right] \geq 0 \Leftrightarrow$$
$$\frac{L_{\max}\rho_t}{2} + BL_{\max}\rho_t(1-\lambda_t)^2 + BL_{\max}\gamma_t \left( 2L_{\max}^2 \rho_t^2(1-\lambda_t)^2 + 1 \right) \leq 1$$

Solving for $\gamma_t$ we get

$$\gamma_t \leq \frac{2 - L_{\max}\rho_t - 2BL_{\max}\rho_t(1-\lambda_t)^2}{2BL_{\max} \left( 2L_{\max}^2 \rho_t^2(1-\lambda_t)^2 + 1 \right)}. \tag{10}$$

The numerator of the upper bound on $\gamma_t$ in (10) must be positive (positive step-size). For this, the following restriction on $\rho_t$ is needed.

$$2 - L_{\max}\rho_t - 2BL_{\max}\rho_t(1-\lambda_t)^2 \geq 0 \Leftrightarrow \rho_t \leq \frac{2}{L_{\max} \left( 1 + 2B(1-\lambda_t)^2 \right)}. \tag{11}$$

Now using the $\mu$-PL condition (3) on (9) we get

$$\mathbb{E}[f(x^{t+1}) - f(x^*)|x^t] - [f(x^t) - f(x^*)]$$
$$\leq -2\gamma_t \mu \left( 1 - \frac{L_{\max}\rho_t}{2} - BL_{\max} \left[ \rho_t(1-\lambda_t)^2 + 2\gamma_t L_{\max}^2 \rho_t^2(1-\lambda_t)^2 + \gamma_t \right] \right) [f(x^t) - f(x^*)]$$
$$+ 2A\gamma_t L_{\max} \left[ \rho_t(1-\lambda_t)^2 + 2\gamma_t L_{\max}^2 \rho_t^2(1-\lambda_t)^2 + \gamma_t \right] \left[ f(x^t) - f(x^*) \right]$$
$$+ \gamma_t L_{\max} \left[ C\rho_t(1-\lambda_t)^2 + 2C\gamma_t L_{\max}^2 \rho_t^2(1-\lambda_t)^2 + C\gamma_t + \rho_t \lambda_t^2 + 2\gamma_t L_{\max}^2 \rho_t^2 \lambda_t^2 \right],$$

hence

$$\mathbb{E}[f(x^{t+1}) - f(x^*)|x^t]$$
$$\leq \left( 1 - 2\gamma_t \mu \left( 1 - \frac{L_{\max}\rho_t}{2} - BL_{\max} \left[ \rho_t(1-\lambda_t)^2 + 2\gamma_t L_{\max}^2 \rho_t^2(1-\lambda_t)^2 + \gamma_t \right] \right) \right.$$
$$\left. + 2A\gamma_t L_{\max} \left[ \rho_t(1-\lambda_t)^2 + 2\gamma_t L_{\max}^2 \rho_t^2(1-\lambda_t)^2 + \gamma_t \right] \right) \left[ f(x^t) - f(x^*) \right]$$
$$+ \gamma_t L_{\max} \left[ C\rho_t(1-\lambda_t)^2 + 2C\gamma_t L_{\max}^2 \rho_t^2(1-\lambda_t)^2 + C\gamma_t + \rho_t \lambda_t^2 + 2\gamma_t L_{\max}^2 \rho_t^2 \lambda_t^2 \right]. \tag{12}$$

The coefficient of $[f(x^t) - f(x^*)]$ can be simplified to

$$1 - 2\gamma_t \mu \left( 1 - \frac{L_{\max}\rho_t}{2} - BL_{\max} \left[ \rho_t(1-\lambda_t)^2 + 2\gamma_t L_{\max}^2 \rho_t^2(1-\lambda_t)^2 + \gamma_t \right] \right)$$
$$+ 2A\gamma_t L_{\max} \left[ \rho_t(1-\lambda_t)^2 + 2\gamma_t L_{\max}^2 \rho_t^2(1-\lambda_t)^2 + \gamma_t \right]$$
$$= 1 - 2\gamma_t \mu + \gamma_t \mu L_{\max}\rho_t + 2\gamma_t \mu BL_{\max} \left[ \rho_t(1-\lambda_t)^2 + 2\gamma_t L_{\max}^2 \rho_t^2(1-\lambda_t)^2 + \gamma_t \right]$$
$$+ 2A\gamma_t L_{\max} \left[ \rho_t(1-\lambda_t)^2 + 2\gamma_t L_{\max}^2 \rho_t^2(1-\lambda_t)^2 + \gamma_t \right]$$
$$= 1 - 2\gamma_t \mu + \gamma_t \mu L_{\max}\rho_t + 2\gamma_t L_{\max}(B\mu + A) \left[ \rho_t(1-\lambda_t)^2 + 2\gamma_t L_{\max}^2 \rho_t^2(1-\lambda_t)^2 + \gamma_t \right]$$
$$= 1 - 2\gamma_t \mu + \gamma_t L_{\max}\rho_t \left( \mu + 2(B\mu + A)(1-\lambda_t)^2 \right) + 2\gamma_t^2 L_{\max}(B\mu + A) \left( 2L_{\max}^2 \rho_t^2(1-\lambda_t)^2 + 1 \right)$$

Let us bound the above coefficient by $1 - \gamma_t \mu$ (used for the final linear convergence). For doing that, we force the following restrictions for $\rho_t$ and $\gamma_t$:

$$
\begin{aligned}
1 - 2\gamma_t\mu &+ \gamma_t L_{\max}\rho_t \left(\mu + 2(B\mu + A)(1 - \lambda_t)^2\right) \\
&+ 2\gamma_t^2 L_{\max}(B\mu + A)\left(2L_{\max}^2\rho_t^2(1 - \lambda_t)^2 + 1\right) \leq 1 - \gamma_t\mu \Leftrightarrow \\
\gamma_t L_{\max}\rho_t &\left(\mu + 2(B\mu + A)(1 - \lambda_t)^2\right) \\
&+ 2\gamma_t^2 L_{\max}(B\mu + A)\left(2L_{\max}^2\rho_t^2(1 - \lambda_t)^2 + 1\right) \leq \gamma_t\mu \Leftrightarrow \\
L_{\max}\rho_t &\left(\mu + 2(B\mu + A)(1 - \lambda_t)^2\right) \\
&+ 2\gamma_t L_{\max}(B\mu + A)\left(2L_{\max}^2\rho_t^2(1 - \lambda_t)^2 + 1\right) \leq \mu \Leftrightarrow \\
2\gamma_t L_{\max}(B\mu + A) &\left(2L_{\max}^2\rho_t^2(1 - \lambda_t)^2 + 1\right) \leq \mu - L_{\max}\rho_t\left(\mu + 2(B\mu + A)(1 - \lambda_t)^2\right) \Leftrightarrow \\
\gamma_t &\leq \frac{\mu - L_{\max}\rho_t\left(\mu + 2(B\mu + A)(1 - \lambda_t)^2\right)}{2L_{\max}(B\mu + A)\left(2L_{\max}^2\rho_t^2(1 - \lambda_t)^2 + 1\right)}.
\end{aligned} \tag{13}
$$

The numerator of the upper bound on $\gamma_t$ in (13) must be positive (as the step-size is positive). Thus:

$$
\mu - L_{\max}\rho_t\left(\mu + 2(B\mu + A)(1 - \lambda_t)^2\right) \geq 0 \Leftrightarrow \rho_t \leq \frac{\mu}{L_{\max}\left(\mu + 2(B\mu + A)(1 - \lambda_t)^2\right)}. \tag{14}
$$

Now by combining Equations (11) and (14) we obtain:

$$
\begin{aligned}
\rho_t \leq \rho^* &= \min\left\{\frac{2}{L_{\max}\left(1 + 2B(1 - \lambda_t)^2\right)}, \frac{\mu}{L_{\max}\left(\mu + 2(B\mu + A)(1 - \lambda_t)^2\right)}\right\} \\
&= \frac{\mu}{L_{\max}\left(\mu + 2(B\mu + A)(1 - \lambda_t)^2\right)},
\end{aligned}
$$

where last equality holds due to:

$$
\begin{aligned}
\frac{\mu}{L_{\max}\left(\mu + 2(B\mu + A)(1 - \lambda_t)^2\right)} &\leq \frac{2}{L_{\max}\left(1 + 2B(1 - \lambda_t)^2\right)} \Leftrightarrow \\
L_{\max}\mu + 2L_{\max}B\mu(1 - \lambda_t)^2 &\leq 2L_{\max}\mu + 4L_{\max}(B\mu + A)(1 - \lambda_t)^2 \Leftrightarrow \\
0 &\leq L_{\max}\mu + 2L_{\max}(B\mu + 2A)(1 - \lambda_t)^2.
\end{aligned}
$$

Similarly for $\gamma_t$ by combining Equations (13) and (10) we obtain:

$$
\begin{aligned}
\gamma_t \leq \gamma^* &= \min\left\{\frac{2 - L_{\max}\rho_t - 2BL_{\max}\rho_t(1 - \lambda_t)^2}{2BL_{\max}\left(2L_{\max}^2\rho_t^2(1 - \lambda_t)^2 + 1\right)}, \frac{\mu - L_{\max}\rho_t\left(\mu + 2(B\mu + A)(1 - \lambda_t)^2\right)}{2L_{\max}(B\mu + A)\left(2L_{\max}^2\rho_t^2(1 - \lambda_t)^2 + 1\right)}\right\} \\
&= \frac{\mu - L_{\max}\rho_t\left(\mu + 2(B\mu + A)(1 - \lambda_t)^2\right)}{2L_{\max}(B\mu + A)\left(2L_{\max}^2\rho_t^2(1 - \lambda_t)^2 + 1\right)},
\end{aligned}
$$

where the last inequality holds due to:

$$
\begin{aligned}
\frac{\mu - L_{\max}\rho_t\left(\mu + 2(B\mu + A)(1 - \lambda_t)^2\right)}{2L_{\max}(B\mu + A)\left(2L_{\max}^2\rho_t^2(1 - \lambda_t)^2 + 1\right)} &\leq \frac{2 - L_{\max}\rho_t - 2BL_{\max}\rho_t(1 - \lambda_t)^2}{2BL_{\max}\left(2L_{\max}^2\rho_t^2(1 - \lambda_t)^2 + 1\right)} \Leftrightarrow \\
\frac{\mu - L_{\max}\rho_t\left(\mu + 2(B\mu + A)(1 - \lambda_t)^2\right)}{B\mu + A} &\leq \frac{2 - L_{\max}\rho_t - 2BL_{\max}\rho_t(1 - \lambda_t)^2}{B} \Leftrightarrow \\
B\mu - BL_{\max}\rho_t\mu - 2BL_{\max}\rho_t(B\mu + A)(1 - \lambda_t)^2 &\leq 2B\mu + 2A - B_{\max}\rho_t\mu - AL_{\max}\rho_t \\
&\quad - 2BL_{\max}\rho_t(B\mu + A)(1 - \lambda_t)^2,
\end{aligned}
$$

The above derivation simplifies to the below inequality:

$$
0 \leq B\mu + A(2 - L_{\max}\rho_t),
$$

which is true by the restriction on $\rho$ in (11), that implies the quantity $(2 - L_{\max}\rho_t)$ is non-negative due to:

$$\rho_t \leq \frac{2}{L_{\max}\left(1 + 2B(1 - \lambda_t)^2\right)} \leq \frac{2}{L_{\max}}.$$

Now setting constant $\rho_t = \rho^*$, $\gamma_t = \gamma^*$ and $\lambda_t = \lambda \in [0, 1]$ in (12) we have

$$\mathbb{E}[f(x^{t+1}) - f(x^*)|x^t]$$
$$\leq (1 - \gamma\mu)[f(x^t) - f(x^*)]$$
$$+ \gamma L_{\max}\left[C\rho(1 - \lambda)^2 + 2C\gamma L_{\max}^2\rho^2(1 - \lambda)^2 + C\gamma + \rho\lambda^2 + 2\gamma L_{\max}^2\rho^2\lambda^2\right]. \quad (15)$$

Taking expectation on (15) and using the tower property we get

$$\mathbb{E}[f(x^{t+1}) - f(x^*)]$$
$$\leq (1 - \gamma\mu)\,\mathbb{E}[f(x^t) - f(x^*)]$$
$$+ \gamma L_{\max}\left[C\rho(1 - \lambda)^2 + 2C\gamma L_{\max}^2\rho^2(1 - \lambda_t)^2 + C\gamma + \rho\lambda^2 + 2\gamma L_{\max}^2\rho^2\lambda^2\right]$$
$$= (1 - \gamma\mu)\,\mathbb{E}[f(x^t) - f(x^*)] + \gamma L_{\max}\left(C\gamma + \rho(1 + 2\gamma L_{\max}^2\rho)\left[\lambda^2 + C(1 - \lambda_t)^2\right]\right). \quad (16)$$

Recursively applying the above and summing up the resulting geometric series yields:

$$\mathbb{E}[f(x^{t+1}) - f(x^*)] \leq (1 - \gamma\mu)^t\left[f(x^0) - f(x^*)\right]$$
$$+ \gamma L_{\max}\left(C\gamma + \rho(1 + 2\gamma L_{\max}^2\rho)\left[\lambda^2 + C(1 - \lambda)^2\right]\right)\sum_{j=0}^{t}(1 - \gamma\mu)^j.$$

Using the fact that $\sum_{j=0}^{t}(1 - \gamma\mu)^j \leq \frac{1}{\gamma\mu}$ completes the proof. $\qquad\square$

### C.2 PROOF OF THEOREM 3.5

**Theorem C.2.** *Assume that each $f_i$ is $L_i$-smooth, $f$ is $\mu$-PL and the ER is satisfied. Then the iterates of Unified SAM with $\rho_t = \min\left\{\frac{1}{2t+1}, \rho^*\right\}$ and $\gamma_t = \min\left\{\frac{2t+1}{(t+1)^2\mu}, \gamma^*\right\}$, where $\rho^*$ and $\gamma^*$ are defined in Theorem 3.2, satisfy:*

$$E[f(x^t) - f(x^*)] \leq O\left(\frac{1}{t}\right).$$

*Proof of Theorem 3.5.* Since $\gamma_t = \min\left\{\frac{2t+1}{(t+1)^2\mu}, \gamma^*\right\} \leq \gamma^*$ and $\rho_t = \min\left\{\frac{1}{2t+1}, \rho^*\right\} \leq \rho^*$ we get that inequality (16) holds for any $t \geq 0$, hence we have:

$$\mathbb{E}[f(x^{t+1}) - f(x^*)] \leq (1 - \gamma_t\mu)\,\mathbb{E}[f(x^t) - f(x^*)]$$
$$+ \gamma_t L_{\max}\left(C\gamma_t + \rho_t(1 + 2\gamma_t\rho_t L_{\max}^2)\left[\lambda^2 + C(1 - \lambda)^2\right]\right)$$
$$= (1 - \gamma_t\mu)\,\mathbb{E}[f(x^t) - f(x^*)]$$
$$+ CL_{\max}\gamma_t^2 + \gamma_t\rho_t L_{\max}^2(1 + 2\gamma_t\rho_t L_{\max}^2)\left[\lambda^2 + C(1 - \lambda)^2\right]$$
$$\leq (1 - \gamma_t\mu)\,\mathbb{E}[f(x^t) - f(x^*)]$$
$$+ CL_{\max}\gamma_t^2 + \gamma_t\rho_t L_{\max}^2\left(1 + \frac{2L_{\max}^2}{\mu}\right)\left[\lambda^2 + C(1 - \lambda)^2\right], \quad (17)$$

where the last inequality follows from $\gamma_t \leq \frac{2t+1}{(t+1)^2\mu}$ and $\rho_t \leq \frac{1}{2t+1}$, and thus $\gamma_t\rho_t \leq \frac{1}{(t+1)^2\mu} \leq \frac{1}{\mu}$. Now set $\delta_t = \mathbb{E}[f(x^t) - f(x^*)]$, $R = CL_{\max}$ and $Q = L_{\max}^2\left(1 + \frac{2L_{\max}^2}{\mu}\right)\left[\lambda^2 + C(1 - \lambda)^2\right]$. Then the inequality (17) takes the following form:

$$\delta_{t+1} \leq (1 - \gamma_t\mu)\delta_t + Q\gamma_t\rho_t + R\gamma_t^2. \quad (18)$$

Now since the sequences $\frac{2t+1}{(t+1)^2\mu}$ and $\frac{1}{2t+1}$ are decreasing there exists an integer $t^* \in \mathbb{N}$ such that for any $t \geq t^*$ we have $\gamma_t = \frac{2t+1}{(t+1)^2\mu}$ and $\rho_t = \frac{1}{2t+1}$. Substituting in (18) we get that for any $t \geq t^*$ we have

$$\delta_{t+1} \leq \frac{t^2}{(t+1)^2}\delta_t + \frac{Q}{\mu(t+1)^2} + \frac{R(2t+1)^2}{\mu^2(t+1)^4}$$
$$\leq \frac{t^2}{(t+1)^2}\delta_t + \frac{Q\mu + 4R}{\mu^2(t+1)^2},$$

because $\frac{(2t+1)^2}{(t+1)^4} \leq \frac{4(t+1)^2}{(t+1)^4} = \frac{4}{(t+1)^2}$. Multiplying both sides with $(t+1)^2$ and rearranging we have

$$(t+1)^2\delta_{t+1} - t^2\delta_t \leq \frac{Q\mu + 4R}{\mu^2}.$$

Summing for $t = t^*, \ldots, T-1$ and telescoping we have

$$T^2\delta_T \leq (t^*)^2\delta_{t^*} + \frac{Q\mu + 4R}{\mu^2}(T - t^* - 1).$$

Changing notation from $T$ to $t$ we get

$$\mathbb{E}[f(x^t) - f(x^*)] \leq \frac{(t^*)^2\delta_{t^*}}{t^2} + \frac{Q\mu + 4R}{\mu^2}\frac{t - t^* - 1}{t^2}$$
$$\leq \frac{(t^*)^2\delta_{t^*}}{t^2} + \frac{Q\mu + 4R}{\mu^2}\frac{1}{t}$$
$$= O\left(\frac{1}{t}\right).$$

This completes the proof. $\qquad\square$

## C.3 PROOF OF PROPOSITION 3.6

**Proposition C.3.** *Assume that each $f_i$ is $L_i$-smooth and the ER is satisfied. Then the iterates of Unified SAM with $\rho \leq \min\left\{\frac{1}{4L_{\max}}, \frac{1}{8BL_{\max}(1-\lambda)^2}\right\}$ and $\gamma \leq \frac{1}{8BL_{\max}}$, satisfy:*

$$\min_{t=0,\ldots,T-1}\mathbb{E}\|\nabla f(x^t)\|^2 \leq \frac{2\left(1 + 2A\gamma L_{\max}\left[\rho(1-\lambda)^2(1 + 2\gamma\rho L_{\max}^2) + \gamma\right]\right)^T}{T\gamma}[f(x^0) - f^{\inf}]$$
$$+ 2L_{\max}\left[C\gamma + \rho(1 + 2\gamma\rho L_{\max}^2)(\lambda^2 + C(1-\lambda)^2)\right].$$

*Proof of Proposition 3.6.* From Equation (9) we have

$$\mathbb{E}[f(x^{t+1}) - f^{\inf}|x^t] - [f(x^t) - f^{\inf}]$$
$$\leq -\gamma\left(1 - \frac{L_{\max}\rho}{2} - BL_{\max}\left[\rho(1-\lambda)^2 + 2\gamma L_{\max}^2\rho^2(1-\lambda)^2 + \gamma\right]\right)\|\nabla f(x^t)\|^2$$
$$+ 2A\gamma L_{\max}\left[\rho(1-\lambda)^2 + 2\gamma L_{\max}^2\rho^2(1-\lambda)^2 + \gamma\right][f(x^t) - f^{\inf}]$$
$$+ \gamma L_{\max}\left[C\rho(1-\lambda)^2 + 2C\gamma L_{\max}^2\rho^2(1-\lambda)^2 + C\gamma + \rho\lambda^2 + 2\gamma L_{\max}^2\rho^2\lambda^2\right]$$
$$\leq -\frac{\gamma}{2}\|\nabla f(x^t)\|^2 + 2A\gamma L_{\max}\left[\rho(1-\lambda)^2 + 2\gamma L_{\max}^2\rho^2(1-\lambda)^2 + \gamma\right][f(x^t) - f^{\inf}]$$
$$+ \gamma L_{\max}\left[C\rho(1-\lambda)^2 + 2C\gamma L_{\max}^2\rho^2(1-\lambda)^2 + C\gamma + \rho\lambda^2 + 2\gamma L_{\max}^2\rho^2\lambda^2\right]. \qquad (19)$$

Here inequality (19) is true due to:

$$\left(1 - \frac{L_{\max}\rho}{2} - BL_{\max}\left[\rho(1-\lambda)^2 + 2\gamma L_{\max}^2\rho^2(1-\lambda)^2 + \gamma\right]\right) \geq \frac{1}{2} \Leftrightarrow$$
$$\frac{L_{\max}\rho}{2} + BL_{\max}\rho(1-\lambda)^2 + BL_{\max}\gamma + 2B\gamma L_{\max}^3\rho^2(1-\lambda)^2 \leq \frac{1}{2}. \qquad (20)$$

Inequality (20) holds due to the theorem's constraints on $\gamma$ and $\rho$, which imply that each factor on the left-hand side of inequality (20) is less than 1/8, as explained below.

- $\frac{L_{\max}\rho}{2} \le \frac{1}{8}$ by the condition on $\rho$.

- $BL_{\max}\rho(1-\lambda)^2 \le \frac{1}{8}$ by the condition on $\rho$.

- $BL_{\max}\gamma \le \frac{1}{8}$ by the condition on $\gamma$.

- $2B\gamma L_{\max}^3\rho^2(1-\lambda)^2 \le \frac{1}{8}$ because we have $\rho \le \frac{1}{4L_{\max}}$ and $\gamma \le \frac{1}{8BL_{\max}}$, so

$$2B\gamma L_{\max}^3\rho^2(1-\lambda)^2 \le 2B\frac{1}{8BL_{\max}}L_{\max}^3\left(\frac{1}{4L_{\max}}\right)^2(1-\lambda)^2$$
$$= \frac{(1-\lambda)^2}{64}$$
$$\le \frac{1}{64} < \frac{1}{8}$$

Summing up the inequalities in the above bullet points results in inequality (20). By taking expectation and using the tower property in (19), we have

$$\mathbb{E}[f(x^{t+1}) - f^{\inf}] - \mathbb{E}[f(x^t) - f^{\inf}]$$
$$\le -\frac{\gamma}{2}\,\mathbb{E}\,\|\nabla f(x^t)\|^2 + 2A\gamma L_{\max}\left[\rho(1-\lambda)^2 + 2\gamma L_{\max}^2\rho^2(1-\lambda)^2 + \gamma\right]\mathbb{E}\left[f(x^t) - f^{\inf}\right]$$
$$+ \gamma L_{\max}\left[C\rho(1-\lambda)^2 + 2C\gamma L_{\max}^2\rho^2(1-\lambda)^2 + C\gamma + \rho\lambda^2 + 2\gamma L_{\max}^2\rho^2\lambda^2\right],$$

thus

$$\frac{\gamma}{2}\,\mathbb{E}\,\|\nabla f(x^t)\|^2 \le \left(1 + 2A\gamma L_{\max}\left[\rho(1-\lambda)^2 + 2\gamma L_{\max}^2\rho^2(1-\lambda)^2 + \gamma\right]\right)\mathbb{E}[f(x^t) - f^{\inf}]$$
$$- \mathbb{E}[f(x^{t+1}) - f^{\inf}]$$
$$+ \gamma L_{\max}\left[C\rho(1-\lambda)^2 + 2C\gamma L_{\max}^2\rho^2(1-\lambda)^2 + C\gamma + \rho\lambda^2 + 2\gamma L_{\max}^2\rho^2\lambda^2\right]. \tag{21}$$

Let us now set:

$$\delta_t = \frac{2}{\gamma}\,\mathbb{E}[f(x^t) - f^{\inf}] \ge 0$$
$$r_t = \mathbb{E}\,\|\nabla f(x^t)\|^2 \ge 0$$
$$g = \left(1 + 2A\gamma L_{\max}\left[\rho(1-\lambda)^2 + 2\gamma L_{\max}^2\rho^2(1-\lambda)^2 + \gamma\right]\right)$$
$$= \left(1 + 2A\gamma L_{\max}\left[\gamma + \rho(1-\lambda)^2\left(1 + 2\gamma L_{\max}^2\right)\right]\right) > 1$$
$$N = 2L_{\max}\left[C\rho(1-\lambda)^2 + 2C\gamma L_{\max}^2\rho^2(1-\lambda)^2 + C\gamma + \rho\lambda^2 + 2\gamma L_{\max}^2\rho^2\lambda^2\right]$$
$$= 2L_{\max}\left[C\gamma + \rho(1 + 2\gamma\rho L_{\max}^2)(\lambda^2 + C(1-\lambda)^2)\right] \ge 0$$

and inequality (21) takes the following form:

$$\frac{\gamma}{2}r_t \le g\delta_t - \delta_{t+1} + N. \tag{22}$$

Applying Lemma A.8 to (22) completes the proof. □

## C.4 PROOF OF THEOREM 3.7

**Theorem C.4.** *Let $\varepsilon > 0$ and set $\delta_0 = f(x_0) - f^{\inf} \ge 0$. For*

$$\rho = \min\left\{\frac{1}{4L_{\max}}, \frac{1}{8BL_{\max}(1-\lambda)^2}, \frac{1}{\sqrt{T}}, \frac{\varepsilon^2}{12L_{\max}(C(1-\lambda)^2 + \lambda^2)}\right\}$$

*and*

$$\gamma = \min\left\{\frac{1}{8BL_{\max}}, \frac{1}{2L(1-\lambda)\sqrt{3AL_{\max}}}, \frac{1}{6L_{\max}A(1-\lambda)^2\sqrt{T}}, \frac{1}{\sqrt{6ALT}},\right.$$

$$\frac{\varepsilon^2}{24L_{\max}^3\left(C(1-\lambda)^2+\lambda^2\right)}, \frac{\varepsilon^2}{12L_{\max}C}\Bigg\}$$

*Then provided that*

$$T \geq \frac{\delta_0 L_{\max}}{\varepsilon^2}\max\left\{96B, 24(1-\lambda)\sqrt{3L_{\max}A}, \frac{5184L_{\max}A^2(1-\lambda)^4\delta_0}{\varepsilon^2}, \frac{864\delta_0 A}{\varepsilon^2}, \frac{144C}{\varepsilon^2},\right.$$
$$\left.\frac{288L_{\max}^2(1-\lambda)^2}{\varepsilon^2}\right\}$$

*we have* $\min_{t=0,\ldots,T-1}\mathbb{E}\left\|\nabla f(x^t)\right\| \leq \varepsilon$.

*Proof of Theorem 3.7.* From Proposition C.3 under the condition that $\rho \leq \min\left\{\frac{1}{4L_{\max}}, \frac{1}{8BL_{\max}(1-\lambda)^2}\right\}$ and $\gamma \leq \frac{1}{8BL_{\max}}$ we have

$$\min_{t=0,\ldots,T-1}\mathbb{E}\left\|\nabla f(x^t)\right\|^2 \leq \frac{2\left(1+2A\gamma L_{\max}\left[\rho(1-\lambda)^2(1+2\gamma\rho L_{\max}^2)+\gamma\right]\right)^T}{T\gamma}[f(x^0)-f^{\inf}]$$
$$+ 2L_{\max}\left[C\gamma+\rho(1+2\gamma\rho L_{\max}^2)(\lambda^2+C(1-\lambda)^2)\right]. \tag{23}$$

Using the fact that $1+x \leq \exp(x)$, we have that

$$\left(1+2A\gamma L_{\max}\left[\rho(1-\lambda)^2(1+2\gamma\rho L_{\max}^2)+\gamma\right]\right)^T$$
$$\leq \exp\left(2TA\gamma L_{\max}\left[\rho(1-\lambda)^2(1+2\gamma\rho L_{\max}^2)+\gamma\right]\right)$$
$$\leq \exp(1) < 3. \tag{24}$$

We note that for the second inequality to hold, it is enough to assume that
$$2TA\gamma L_{\max}\left[\rho(1-\lambda)^2(1+2\gamma\rho L_{\max}^2)+\gamma\right] = 2TAL_{\max}\gamma\left[\rho(1-\lambda)^2+2\gamma L_{\max}^2\rho^2(1-\lambda)^2+\gamma\right]$$
$$\leq 1,$$

which is true if we have that,

- $2TAL_{\max}\gamma\rho(1-\lambda)^2 \leq 1/3$

- $4TAL_{\max}^3\gamma\rho^2(1-\lambda)^2 \leq 1/3$

- $2TAL_{\max}\gamma^2 \leq 1/3$.

Imposing the restriction $\rho \leq \frac{1}{\sqrt{T}}$, the above inequalities can be expressed as (by solving for $\gamma$):

- $\gamma \leq \frac{1}{6L_{\max}A(1-\lambda)^2\sqrt{T}}$

- $\gamma \leq \frac{1}{2L(1-\lambda)\sqrt{3AL_{\max}}}$

- $\gamma \leq \frac{1}{\sqrt{6ALT}}$

Using these restrictions on $\gamma$, we are able to substitute the bound of (24) in (23) to get:

$$\min_{t=0,\ldots,T-1}\mathbb{E}\left\|\nabla f(x^t)\right\|^2 \leq \frac{6\delta_0}{T\gamma} + 2L_{\max}\left[C\gamma+\rho(1+2\gamma\rho L_{\max}^2)(\lambda^2+C(1-\lambda)^2)\right]. \tag{25}$$

To make the right hand side of (25) smaller than $\varepsilon^2$, we require that the second term satisfies

$$2L_{\max}\left[C\gamma+\rho(1+2\gamma\rho L_{\max}^2)(\lambda^2+C(1-\lambda)^2)\right] \leq \frac{\varepsilon^2}{2} \Leftrightarrow$$

$$2L_{\max}\left[C\rho(1-\lambda)^2+2C\gamma L_{\max}^2\rho^2(1-\lambda)^2+C\gamma+\rho\lambda^2+2\gamma L_{\max}^2\rho^2\lambda^2\right] \leq \frac{\varepsilon^2}{2} \Leftrightarrow$$

$$2L_{\max}\left[\rho\left(C(1-\lambda)^2+\lambda^2\right)+2L_{\max}^2\gamma\rho^2\left(C(1-\lambda)^2+\lambda^2\right)+C\gamma\right] \leq \frac{\varepsilon^2}{2}$$

For this to hold it is enough to have

- $2L_{\max}\rho(C(1-\lambda)^2 + \lambda^2) \leq \frac{\varepsilon^2}{6} \Leftarrow \rho \leq \frac{\varepsilon^2}{12L_{\max}(C(1-\lambda)^2+\lambda^2)}$

- $4L_{\max}^3\gamma\rho^2\left(C(1-\lambda)^2+\lambda^2\right) \leq \frac{\varepsilon^2}{6} \overset{\rho \leq 1}{\Leftarrow} \gamma \leq \frac{\varepsilon^2}{24L_{\max}^3(C(1-\lambda)^2+\lambda^2)}$

- $2L_{\max}C\gamma \leq \frac{\varepsilon^2}{6} \Leftarrow \gamma \leq \frac{\varepsilon^2}{12L_{\max}C}$

Similarly, for the first term, we get that the number of iterations must satisfy:

$$\frac{6\delta_0}{T\gamma} \leq \frac{\varepsilon^2}{2} \Leftrightarrow T \geq \frac{12\delta_0}{\gamma\varepsilon^2} \tag{26}$$

Hence so far we need the following restrictions on $\rho$ and $\gamma$:

- $\rho \leq \min\left\{\frac{1}{4L_{\max}}, \frac{1}{8BL_{\max}(1-\lambda)^2}\right\}$ and $\gamma \leq \frac{1}{8BL_{\max}}$ (by the restrictions of Proposition C.3)

- $\rho \leq \frac{1}{\sqrt{T}}$ and $\rho \leq \frac{\varepsilon^2}{12L_{\max}(C(1-\lambda)^2+\lambda^2)}$

- $\gamma \leq \frac{1}{6L_{\max}A(1-\lambda)^2\sqrt{T}}$ and $\gamma \leq \frac{1}{2L(1-\lambda)\sqrt{3AL_{\max}}}$ and $\gamma \leq \frac{1}{\sqrt{6ALT}}$ and $\gamma \leq \frac{\varepsilon^2}{24L_{\max}^3(C(1-\lambda)^2+\lambda^2)}$ and $\gamma \leq \frac{\varepsilon^2}{12L_{\max}C}$

Plugging each of the previous bounds of $\gamma$ into (26) we get

- $T \geq \frac{96BL_{\max}\delta_0}{\varepsilon^2}$

- $T \geq \frac{24L_{\max}\delta_0(1-\lambda)\sqrt{3L_{\max}A}}{\varepsilon^2}$

- $T \geq \frac{5184L_{\max}^2A^2(1-\lambda)^4\delta_0^2}{\varepsilon^4}$

- $T \geq \frac{864\delta_0^2AL_{\max}}{\varepsilon^4}$

- $T \geq \frac{144L_{\max}C\delta_0}{\varepsilon^4}$

- $T \geq \frac{288L_{\max}^3\delta_0(1-\lambda)^2}{\varepsilon^4}$

Finally, collecting all the terms into a single bound we have:

$$\rho = \min\left\{\frac{1}{4L_{\max}}, \frac{1}{8BL_{\max}(1-\lambda)^2}, \frac{1}{\sqrt{T}}, \frac{\varepsilon^2}{12L_{\max}(C(1-\lambda)^2+\lambda^2)}\right\}$$
$$\gamma = \min\left\{\frac{1}{8BL_{\max}}, \frac{1}{2L(1-\lambda)\sqrt{3AL_{\max}}}, \frac{1}{6L_{\max}A(1-\lambda)^2\sqrt{T}}, \frac{1}{\sqrt{6ALT}},\right.$$
$$\left.\frac{\varepsilon^2}{24L_{\max}^3(C(1-\lambda)^2+\lambda^2)}, \frac{\varepsilon^2}{12L_{\max}C}\right\}$$
$$T \geq \frac{\delta_0 L_{\max}}{\varepsilon^2}\max\left\{96B, 24(1-\lambda)\sqrt{3L_{\max}A}, \frac{5184L_{\max}A^2(1-\lambda)^4\delta_0}{\varepsilon^2}, \frac{864\delta_0A}{\varepsilon^2}, \frac{144C}{\varepsilon^2},\right.$$
$$\left.\frac{288L_{\max}^2(1-\lambda)^2}{\varepsilon^2}\right\}.$$

This completes the proof. $\qquad\square$

## D  ADDITIONAL EXPERIMENTS

In this section, we present additional experimental evaluations of Unified SAM, following the same setup as in Li & Giannakis (2023). Specifically, we train ResNet-18 and WRN-28-10 on CIFAR10 and CIFAR100 datasets. Standard data augmentation techniques, including random cropping, random horizontal flipping, and normalization DeVries (2017), are employed. The models are trained for 200 epochs with a batch size of 128, using a cosine scheduler starting from 0.05. Weight decay is set to 0.001. For SAM, we use $\rho = 0.1$ for CIFAR10 and $\rho = 0.2$ for CIFAR100. Each experiment is repeated three times, and we report the average of the maximum test accuracy along with the standard error. The numerical results are presented in Tables 6 and 7, demonstrate that Unified SAM consistently improves the test accuracy over both USAM and SAM across all tested models. Lastly, we also observe that careful tuning of the parameter $\lambda$ is essential for achieving optimal performance.

Table 6: Test accuracy (%) of Unified SAM on various neural networks trained on CIFAR10.

| Model | $\lambda = 0.0$ | $\lambda = 0.5$ | $\lambda = 1.0$ | $\lambda = 1/t$ | $\lambda = 1 - 1/t$ |
|---|---|---|---|---|---|
| ResNet-18 | $96.13_{\pm 0.05}$ | $96.16_{\pm 0.03}$ | $\mathbf{96.33}_{\pm 0.03}$ | $96.20_{\pm 0.05}$ | $96.22_{\pm 0.09}$ |
| WideResNet-28-10 | $\mathbf{97.26}_{\pm 0.44}$ | $96.98_{\pm 0.08}$ | $97.05_{\pm 0.05}$ | $96.73_{\pm 0.04}$ | $96.63_{\pm 0.35}$ |

Table 7: Test accuracy (%) of Unified SAM on various neural networks trained on CIFAR100.

| Model | $\lambda = 0.0$ | $\lambda = 0.5$ | $\lambda = 1.0$ | $\lambda = 1/t$ | $\lambda = 1 - 1/t$ |
|---|---|---|---|---|---|
| ResNet-18 | $80.21_{\pm 0.02}$ | $\mathbf{80.28}_{\pm 0.16}$ | $80.14_{\pm 0.07}$ | $80.27_{\pm 0.09}$ | $80.19_{\pm 0.06}$ |
| WideResNet-28-10 | $83.49_{\pm 0.23}$ | $\mathbf{83.71}_{\pm 0.03}$ | $83.55_{\pm 0.19}$ | $83.55_{\pm 0.10}$ | $83.62_{\pm 0.16}$ |

Table 8: Test accuracy (%) of Unified SAM for PRN-18 on CIFAR10, evaluated across different values of $\rho$ and $\lambda$. With bold we highlight the best performance for fixed $\rho$.

| Unified SAM | $\lambda = 0.0$ | $\lambda = 0.5$ | $\lambda = 1.0$ | $\lambda = 1/t$ | $\lambda = 1 - 1/t$ |
|---|---|---|---|---|---|
| $\rho = 0.1$ | $95.29_{\pm 0.09}$ | $95.32_{\pm 0.02}$ | $\mathbf{95.59}_{\pm 0.09}$ | $95.24_{\pm 0.03}$ | $95.53_{\pm 0.11}$ |
| $\rho = 0.2$ | $95.25_{\pm 0.14}$ | $95.48_{\pm 0.05}$ | $95.5_{\pm 0.02}$ | $95.38_{\pm 0.11}$ | $\mathbf{95.58}_{\pm 0.07}$ |
| $\rho = 0.3$ | $95.25_{\pm 0.11}$ | $95.24_{\pm 0.02}$ | $95.18_{\pm 0.04}$ | $95.12_{\pm 0.19}$ | $\mathbf{95.26}_{\pm 0.1}$ |
| $\rho = 0.4$ | $94.76_{\pm 0.09}$ | $94.98_{\pm 0.07}$ | $\mathbf{94.7}_{\pm 0.02}$ | $94.64_{\pm 0.03}$ | $94.61_{\pm 0.09}$ |
| SGD | | | $94.82_{\pm 0.02}$ | | |

Table 9: Test accuracy (%) of Unified SAM for PRN-18 on CIFAR100, evaluated across different values of $\rho$ and $\lambda$. With bold we highlight the best performance for fixed $\rho$.

| Unified SAM | $\lambda = 0.0$ | $\lambda = 0.5$ | $\lambda = 1.0$ | $\lambda = 1/t$ | $\lambda = 1 - 1/t$ |
|---|---|---|---|---|---|
| $\rho = 0.1$ | $78.28_{\pm 0.15}$ | $78.28_{\pm 0.06}$ | $78.32_{\pm 0.22}$ | $78.33_{\pm 0.32}$ | $\mathbf{78.39}_{\pm 0.31}$ |
| $\rho = 0.2$ | $\mathbf{78.98}_{\pm 0.18}$ | $78.68_{\pm 0.13}$ | $78.96_{\pm 0.12}$ | $78.87_{\pm 0.02}$ | $78.79_{\pm 0.1}$ |
| $\rho = 0.3$ | $79.0_{\pm 0.05}$ | $78.95_{\pm 0.07}$ | $79.21_{\pm 0.08}$ | $78.73_{\pm 0.06}$ | $\mathbf{79.27}_{\pm 0.08}$ |
| $\rho = 0.4$ | $78.57_{\pm 0.26}$ | $78.76_{\pm 0.3}$ | $\mathbf{79.05}_{\pm 0.16}$ | $78.36_{\pm 0.09}$ | $78.79_{\pm 0.13}$ |
| SGD | | | $76.9_{\pm 0.23}$ | | |

