# OpenReview forum: "Sharpness-Aware Minimization: General Analysis and Improved Rates"
_ICLR.cc/2025/Conference — ICLR 2025 Poster_

### Official Review · Reviewer_hnag · 2024-10-29

**Soundness:** 2
**Presentation:** 3
**Contribution:** 2
**Rating:** 5
**Confidence:** 4

**Summary:**

The paper studies the convergence of SAM and USAM in stochastic settings. It proves these properties for a newly proposed algorithm, Unified-SAM, which includes SAM and USAM as special cases. The analysis relaxes popular assumptions like bounded variance (BV) and gradients (BG), replacing them with expected residual (ER) condition. The proof provides convergence guarantees for SAM under different step sizes in non-convex functions and Polyak-Lojasiewicz (PL) functions. The theory holds under arbitrary sampling paradigms, including importance sampling. The authors also demonstrate Unified-SAM's performance compared to SAM in practical settings.

**Strengths:**

- The paper is well-written and easy to follow.
- The authors challenge existing assumptions on stochastic noise, such as bounded gradients and bounded variance, in proving SAM's convergence, replacing them with an expected residual condition that encompasses both as special cases.
- The proof is a slight improvement over existing studies.

**Weaknesses:**

- The empirical results from the original SAM paper Foret et al. [2020] are established using a constant $\rho$. This is a crucial point for aligning theoretical and empirical results. To my knowledge, there are no existing works that establish convergence results on the constant $\rho$ for non-convex functions. However, there are some theoretical papers that use conditions closely related to the constant $\rho$, but these have not been discussed in the paper under review, for example,  Nam et al. [2023], Khanh et al. [2024], Xie et al. [2024]. The assumption in this paper regarding $\rho$ is $\rho = \min ( \frac{1}{2t+1}, \rho^{\star} )$, which is less general than the assumption in Nam et al. [2023], where $\rho$ is defined as $\sum^{\infty}_{t=0} \rho_t^2 < \infty$. Additionally, the assumption on $\rho$ in Khanh et al. [2024] for the full-batch setting is even more general, as it allows the perturbation radius $\rho$ to decrease at arbitrary slow rates, which nearly captures the constant $\rho$.
- In the asymptotic setting, this paper's result in Theorem 3.7, $\min_{t=0,...,T-1} \mathbb{E} || \nabla f(x^t) || \leq \epsilon$, is weaker than the convergence result in the stochastic, non-convex setting in Nam et al. [2023], where the gradient norm approaches zero almost surely. Furthermore, if Theorem 3.7 is considered in the full-batch setting, it is also weaker than the result in Khanh et al. [2024.
- In Theorem 3.5, why do you write $O\left( \frac{1}{t} + \frac{1}{t^2} \right)$ instead of $O\left( \frac{1}{t} \right)$, since they are equivalent?
- Compare your Assumption 3.1 (Expected Residual Condition) with Assumption A.4 in Nam et al. [2023].
- The name of the algorithm, Unified-SAM, may not be suitable, as it only covers USAM, SAM, and the variant that transitions between SAM and USAM. There are many other SAM-like variants that this algorithm does not cover.
- As shown in Tables 2 and 3, the proposed Unified-SAM does not show significant improvement over SAM. This is a point that diminishes the importance and contribution of the paper.
- Line 152-153: “This is the first result that drops the bounded variance assumption for both USAM and SAM." I suggest rewriting this sentence to clarify that, while the bounded variance assumption is removed, an Expected Residual condition is still required.

References:
- Pierre Foret, Ariel Kleiner, Hossein Mobahi, and Behnam Neyshabur. Sharpness-aware minimization
for efficiently improving generalization. ICLR 2021. URL https://arxiv.org/pdf/2010.01412.
- Pham Duy Khanh, Hoang-Chau Luong, Boris S Mordukhovich, and Dat Ba Tran. Fundamental convergence analysis of sharpness-aware minimization. NeurIPS, 2024. URL https://arxiv.org/pdf/2401.08060.
- Kyunghun Nam, Jinseok Chung, and Namhoon Lee. Almost sure last iterate convergence of sharpness-aware minimization. Tiny Papers ICLR, 2023. URL https://openreview.net/forum?id=IcDTYTI0Nx.
- Wanyun Xie, Thomas Pethick, and Volkan Cevher. Sampa: Sharpness-aware minimization parallelized. NeurIPS, 2024. URL https://arxiv.org/pdf/2410.10683v1.

**Questions:**

See the Weakness part.

---

> ### Author Response · Authors · 2024-11-20
> **Authors' response to Reviewer hnag [1/2]**
>
> We would like to thank the reviewer for their time. Below, we address questions and concerns raised by the reviewer.
>
> * We agree with the reviewer that Foret et al. [2020] are established using a constant $\rho$. However, we politely disagree with the comment that “no existing works that establish convergence results on the constant $\rho$ for non-convex functions.” As we mentioned in our paper, there are some existing results about constant step-size SAM in the general non-convex setting.
> For example, in [Andriushchenko, 2022] and [Li & Giannakis, 2023] the authors prove convergence of USAM and SAM respectively, with constant stepsizes that depend on $T$ (total number of iterations) with rate $O(1/\sqrt{T})$. Both papers make the strong assumption of bounded variance. All of the above papers include limitations in their convergence guarantees and this is precisely what our work fixes. We provide tight convergence analysis of both USAM and SAM under relaxed assumptions (and we did that under the unified SAM framework we proposed).
>
>     Regarding the mentioned papers, let us provide more details below. We have already cited them in the updated version of our work (see pdf).
>
>     **Khanh et al. [2024]**: This work only considers the deterministic setting and all the results are asymptotic. In our paper, we focus on the stochastic regime (which has the deterministic as a special case) and we provide convergence rates in our results.
>
>     **Nam [2023]**: We were not aware of this work. Thank you for bringing it to our attention. As the reviewer mentioned, we make the same main assumption with this work, namely the expected residual assumption. However, there the authors additionally assume bounded gradient of f and the results are only asymptotic and hold almost surely while **they do not prove any convergence rates (do not provide how fast the proposed method is)**. Furthermore, they have no theory for PL functions.
>
>     **Xie [2024]**: We were not aware of this work. Actually, it would be **impossible to compare with it as it was published on arxiv on Oct 14, while the ICLR deadline was Oct 1**. However, thank you for bringing it to our attention. Compared to our work, this paper only focuses on the deterministic regime, arguably a much simpler setting. This is really concurrent work, and our submission should not be judged because it is missing such (impossible to have) comparisons.
> * We politely disagree with the reviewer's comment on the prior works and their relation to our paper.
> In particular, the reviewer's second statement related to prior work is misleading. None of the previous results mentioned provide convergence rates.
>
>     In our work, we provide convergence rates, and thus, the two results are not directly comparable. In Thm 3.7 we show that Unified SAM yields the rate $O(\epsilon^{-4})$ rate for finding a stationary point of nonconvex smooth functions.  Our results are clearly not weaker than Nam [2023] or Khanh [2024] as the reviewer pointed out. These papers only provide asymptotic convergence. Furthermore, as mentioned in the previous point, in Nam [2023] also assumes a bounded gradient of the function on top of the ER condition, making the results impractical as this condition is very restrictive in practical scenarios.
> * It is updated (please see line 296).
> * Our assumption 3.1 and assumption A.4 in Nam et al. [2023] are equivalent.
> * The name Unified SAM is justified since it indeed unifies the analysis of both USAM and SAM. As we never claim that we have a unified theory for all different variants of SAM we believe that the name of the method is okay. We agree with the reviewer that there are many other SAM-like variants that our approach does not cover and we add a sentence related to this in the updated version.
> * The reviewer mentioned: “Tables 2 and 3, the proposed Unified-SAM does not show significant improvement over SAM. This is a point that diminishes the importance and contribution of the paper.”
> We politely disagree with this statement; even if minor improvement in general our results in terms of generalization aligned well with existing state-of-the-art approaches in the literature on improving SAM. Please check the following papers with similar improvements over SAM: [Li, 2023] Table 1,2; and [Kwon, 2021] Table 6. Even the papers the reviewer mentioned have similar improvements to ours (see Xie [2024] Table 1,2).
> * We have updated this sentence to the new PDF file.
>
> References (not mentioned in our paper):
>
> [Kwon, 2021]: ASAM: Adaptive Sharpness-Aware Minimization for Scale-Invariant Learning of Deep Neural Networks

---

> > ### Author Response · Authors · 2024-11-20
> > **Authors' response to Reviewer hnag [2/2]**
> >
> > **Final Comment:**
> >
> > As we point out above the papers mentioned by the reviewer should not undermine the main contributions of our work. even if they are related our convergence guarantees are focused on the stochastic case and through the unified framework we obtain tight analysis for SAM and USAM and their in-between variants. As such we argue that none of the mentioned points are weaknesses of our work but more clarification of how they are related to prior works.
> >
> > **If you agree that we managed to address all issues, please consider raising your mark to support our work. If you believe this is not the case, please let us know so that we have a chance to respond.**

---

> ### Author Response · Authors · 2024-11-26
>
> Dear Reviewer hnag,
>
> In our rebuttal, we explained in more detail that none of the papers mentioned in your review captures the convergence guarantees of our work and their importance in understanding the behavior of stochastic SAM-type methods. They either focus on the arguably much simpler deterministic case or if they have stochastic results, they only have asymptotic convergence (convergence to a minimum but not convergence rates).
>
> One of the papers mentioned became available online only after the ICLR submission deadline (it was impossible to be aware of this).
>
> Please let us know if you have any other concerns.
>
> In our opinion, based on the original points of the reviewer, none of the issues raised justify a score of rejection/borderline rejection.
> **If you agree that we managed to address all issues, please consider raising your mark. If you believe this is not the case, please let us know so we can respond.**
>
> Thanks,
> The authors

---

> > ### Comment · Reviewer_hnag · 2024-11-29
> > **Thank you for your response**
> >
> > Thank you for your response.
> >
> > My first question concerns the assumption about the perturbation radius $\rho$, rather than the step size. Your proof assumes that
> > $\rho$ diminishes to zero, whereas in practical training, $\rho$ is typically kept constant. This difference limits the practical relevance of your proof compared to that of Khanh et al. [2024], who assume a nearly constant perturbation radius. In Table 1, there is no comparison with Nam et al. [2023], whose work is closely related to yours, which makes it not easy to evaluate the contributions of your work.
> >
> > I would slightly increase my score to reflect your attempt to explain why your work is better than that of Nam et al. [2023].

---

> > > ### Author Response · Authors · 2024-11-29
> > >
> > > Thank you for increasing your score.
> > >
> > > In the last message you mentioned: “Your proof assumes that $\rho$ diminishes to zero, whereas in practical training, $\rho$ is typically kept constant.”
> > >
> > > This is not true. In our paper, we have six different theoretical results expressed as Theorem 3.2,  Corollary 3.3, Corollary 3.4, Theorem 3.5.,  Proposition 3.6, and Theorem 3.7.
> > >
> > > In all of the above (except Theorem 3.5, which focuses on PL functions) parameters $\rho$ and $\gamma$ have constant values not decreasing. We agree with the reviewer that practical training, $\rho$ is typically kept constant and this is exactly what our theorems use. Our step-sizes depend on $T$ (total number of iterations) but they are not decreasing. This aligns well with prior work in the analysis of stochastic SAM-type methods [Mi, 2022], [Andriushchenko, 2022] and [Li, 2023].
> > >
> > > Regarding [Khanh et al]: Indeed the condition on $\rho$ of this paper is weaker than ours; however, in their paper $\rho$ needs to go to $0$. Furthermore, all of Khanh's results exclusively focus on the **deterministic regime** while in our case, we focus on stochastic algorithms. The stochastic setting (which is what we focus on) adds another layer of complexity due to the stochastic noise, and hence, as a general rule (almost) all stochastic results are weaker than their deterministic counterparts.
> > >
> > > Regarding [Nam et al]: As explained in our last response we indeed have the same main assumption with [Nam et al], namely the expected residual. However, in [Nam et al] they additionally assume bounded gradient of $f$. Their main result shows *asymptotic convergence almost surely*. In our result we provide **exact rates that hold surely**. Finally, we note that also their $\rho$ goes to $0$.
> > >
> > > Table 1, as we mentioned in the paper, focuses on prior work that has convergence rates in a stochastic setting. Khanh et al. [2024], have only deterministic results, while as we explained in our previous response, the paper of Nam et al. [2023] does not have any convergence rates (only proves convergence without analysis of how fast the method is). None of the two papers are comparable to our analysis, and we intentionally did not include them in the table but only mentioned them after.
> > >
> > > If you agree with our final clarification of our results, we would appreciate increasing your score further to support our work. Ultimately, the reasoning behind raising the score to just 5 is not accurate. We hope you will agree with us on this.
> > >
> > > **References:**
> > >
> > > [Mi, 2022] Peng Mi, Li Shen, Tianhe Ren, Yiyi Zhou, Xiaoshuai Sun, Rongrong Ji, and Dacheng Tao. Make sharpness-aware minimization stronger: A sparsified perturbation approach. In NeurIPS, 2022.
> > >
> > > [Andriushchenko, 2022] Maksym Andriushchenko and Nicolas Flammarion. Towards understanding sharpness-aware minimization. In ICML, 2022
> > >
> > > [Li, 2023] Bingcong Li and Georgios Giannakis. Enhancing sharpness-aware optimization through variance suppression. In NeurIPS, 2023.

---

> > > > ### Author Response · Authors · 2024-12-04
> > > >
> > > > Dear Reviewer **hnag**,
> > > >
> > > > Thank you once again for your thoughtful feedback on our paper.
> > > > We greatly appreciate the time and effort you’ve put into reviewing and considering our responses during the rebuttal phase.
> > > >
> > > > Based on the additional clarifications we provided, we believe that we have addressed your concerns comprehensively. Even if you are unable to provide further responses at this time (not able to add additional comments officially), we hope you will consider raising your score to reflect the details we provided related to our theoretical results (no decreasing \rho) and the importance of our convergence analysis compare to prior works (convergence guarantees in the fully stochastic setting).
> > > >
> > > > We are grateful for your consideration and for helping ensure a fair evaluation process.
> > > > Thanks again for participating in the discussion.

---

### Official Review · Reviewer_78w2 · 2024-11-04

**Soundness:** 2
**Presentation:** 3
**Contribution:** 2
**Rating:** 5
**Confidence:** 4

**Summary:**

This paper studies the convergence of sharpness-aware minimization (SAM) for smooth functions. The authors proposed a new unified notion of normalized SAM and unnormalized SAM by linearly interpolating the perturbed point used to take the gradient for the next update. The main contribution of the paper is to establish convergence rates to first-order stationary points for unified SAM under a more general noise assumption called "Expected Residual (ER) Condition" and arbitrary sampling methods. For non-convex but PL functions, the authors prove the loss value converges in $O(1/\epsilon)$ steps.

**Strengths:**

This paper is overall well written (except some typos) and the presentation of the main result is good and easy to understand. The theoretical results for convergence of SAM under the Expected Residual (ER) Condition is new. The authors also provide numerical evaluation for the new algorithm proposed in this paper, that is, unified SAM with $\lambda_t = 1-1/t$.

**Weaknesses:**

I have concerns both about the correctness and significance about the main result in this paper. Given these concerns, I do not think the current draft meets the bar of ICLR.

**Correctness**:
- In Theorem 3.7, the authors wrote "choose $\rho\le\bar \rho$ and $\gamma\le \bar\gamma$...", the gradient norm will be smaller than $\epsilon$ in at most $1/\epsilon^4$ setps. It is not clear if the authors mean there exists $\rho\le\bar \rho$ and $\gamma\le \bar\gamma$, or for any $\rho\le\bar \rho$ and $\gamma\le \bar\gamma$. The former interpretation make the result trivial because the optimal choice is $\rho=0$ and unified SAM becomes SGD. Howver, i think the later interpretation makes the current results wrong. As a quick sanity check, learning rate $\gamma=0$ means the gradient should not change at all. (see elaboration in the next point below)

- In the formal statement of Theorem 3.7, which is Theorem C.4, the authors indeed interpret the condition as "for any $\rho\le\bar \rho$ and $\gamma\le \bar\gamma$" (line 1100). However, this does not make sense because it does not exclude the case of $\gamma=0$. A direct cause of this could be that the authors forgot to include Equation (13) into the restrictions that they need to satisfy, which says $T\ge \frac{12\delta_0}{\gamma \epsilon^2}$.

- However, I do not think the above issue can be fixed, unless $\rho(1-\lambda)=0$. The authors tried to replicate the analysis by Khaled \&
Richtarik (2020) for SGD to unified SAM, including how to deal with the seemingly exponential explosion. However, in the case of unified SAM, the term before $[f(x^))-f^{inf}]$ in line 1067 is essentially $\frac{(1+\Theta(\gamma))^T}{T\gamma}$ when $\rho(1-\lambda)\neq 0$.  Because $\frac{(1+\Theta(\gamma))^T}{T\gamma} \ge \frac{(1+\Theta(T\gamma)}{T\gamma} = \Theta(1)$, the right-hand side of line 1066 is at least a constant.

**Significance of Theory**: This paper only talks about optimization of SAM, but indeed, SAM is proposed to improve the generalization of SGD. It is not clear to me what the ultimate goal here by studying these convergence bounds of unified SAM. Both from a intuitive sense or from the bounds presented in this paper, SGD (or unified SAM with $\rho=0$) has the best optimization performance. To me the real problem for SAM is how to show they can efficiently minimize the sharpness-aware loss (either the origial notion of maximal loss after certain perturbation, or the hessian-regularized version proposed in Wen et al. (2023) and Bartlett et al. (2023)), rather than viewing them as a tool to minimize the original training loss and analyze that behavior.

**Significance of Experiments**: The performance of the proposed $1-1/t$ schedule for unified SAM does not beat normalized SAM by a margin which is larger than the standard deviation in most experiments. Sometimes unified SAM even has better generalization. I understand this does not contradict with theory because theory does not try to say anything about generalization. However, it is not clear to me if the main benefit of unified SAM is better optimization or generalization.

**References**:

- Wen, Kaiyue, Tengyu Ma, and Zhiyuan Li. "How Sharpness-Aware Minimization Minimizes Sharpness?." The Eleventh International Conference on Learning Representations. 2023.
- Bartlett, Peter L., Philip M. Long, and Olivier Bousquet. "The dynamics of sharpness-aware minimization: Bouncing across ravines and drifting towards wide minima." Journal of Machine Learning Research 24.316 (2023): 1-36.


# Minor comments:
1. line 271. "our results is the first to demonsteate linear convergence in the fully stochastic regime". I would suggest that the authors do not call this linear converge to avoid confusion, when the final bound for loss is still at least constantly large.
2. line 334, "The results in Proposition 3.6 and Theorem 3.7 are tight, as setting ρ= 0 Unified SAM reduces in SGD and these simplify to the step sizes and rates (up to constants) of Theorem 2 and Corollary 1 from Khaled & Richtarik (2020)." The whole point of the analysis is for the regime $\rho\neq 0$. The fact that the analysis in this paper recovers previous result when $\rho=0$ does not indicate the tightness of the result in the main setting when $\rho\neq 0$, especially the dependence on $\rho$.

# Typos:
1. line 224, missing l2 norm on $g(x)$
2. section B in appendix, line 866-890. Missing norm and square over the norm.
3. line 1121. "From Theorem 3.7" should be from "Proposition C.3"

**Questions:**

See weakness.

---

> ### Author Response · Authors · 2024-11-20
> **Authors' response to Reviewer 78w2 [1/2]**
>
> Thank you for investing time in the review process and for the detailed review. Below, we address questions and concerns raised by the reviewer.
>
> The reviewer has concerns about the correctness and significance of our results and claims that because of these issues, the current draft does not meet the bar for ICLR. **We disagree with this judgment, and we provide more details below. In our opinion, all points mentioned as significant limitations of our work are simply clarification points.**
>
>
> **Correctness:**
> The reviewer raised three points related to the correctness of our results.
>
> The first two points are related to the choice of stepsize gamma. That is, the reviewer argues that having $\gamma=0$ makes the theorem useless. We agree with this statement, and we point out here that $\gamma$ as a step-size is always positive by default. This applies to any practical optimization algorithm (including gradient descent and its different variants). If the step-size is zero, then there is no progress of the method (the point $x^k$ of the method remains fixed), and there is no convergence by default.
>
> As such, the two first correctness points the reviewer mentioned are not really issues related to the correctness of our approach. $\gamma$ is not allowed to be zero in any of our convergence guarantees.
>
>
> The third point related to our analysis of SAM in the non-convex setting was indeed a minor issue, and we appreciate that the reviewer pointed it out. As we explained below and in our updated PDF, this has an easy fix as well. In particular, in our previous draft, in Theorem 3.7 we had step-sizes $\rho=O(1)$ and $\gamma=O(1/T)$, which indeed takes care of the exponential explosion term but might not guarantee convergence.
> However, the fix was quite easy (see updated file, we have used red fonts highlighting the changes in the stepsizes). We just needed to choose $\rho=O(1/\sqrt{T})$ and $\gamma=O(1/\sqrt{T})$. We have updated all the statements and proofs in the new draft.
>
> Thank you for catching this. We greatly appreciate the feedback.
>
>
>
> **Significance of Theory:**
>
> We agree with the statement of the reviewer that for SAM, one of the main challenges is how to show the method can efficiently minimize the sharpness-aware loss (either the original notion of maximal loss after a certain perturbation or the hessian-regularized version). However, in our opinion, for any optimization/training algorithm, the generalization and optimization aspects are of the same importance. Theory in terms of (i) generalization performance and minimizing sharpness-aware loss aims to understand which solution the method converges and explain how this leads to better performance in DNNs and (ii) optimization focus on the algorithm's speed to reach this stationary point.
>
> Both are valuable aspects of any training algorithm. In this work, even if we present some generalization results in experiments we primarily focus on the optimization aspect of SAM variants and explain how fast the different variants of SAM are under realistic assumptions (no strong assumptions made in prior works).
>
> There has been plenty of research output in the last couple of years on papers analyzing the convergence guarantees (no generalization) of the SAM-type method. See, for example, [Khanh, 2024], [Li, 2023], [Andriushchenko, 2022].  Our work belongs in this category of papers. Having said that, the two papers mentioned by the reviewer are very interesting, and results about efficiently minimizing sharpness-aware loss are orthogonal directions to our paper. We suspect combining these papers and our work would be an interesting future direction. We will cite them in the camera-ready version and clarify further the difference between optimization and minimization of the sharpness-aware loss.

---

> > ### Author Response · Authors · 2024-11-20
> > **Authors' response to Reviewer 78w2 [2/2]**
> >
> > **Significance of Experiments:**
> >
> > Indeed, the Unified SAM (with different $\lambda$) beat SAM by a small margin. In practice this is typically the case of most papers in comparison of performance of SAM-type variants Please see for example: [Xie, 2024] Table: 1,2, [Li, 2023] Table 1,2; [Kwon, 2021] Table 6.
> >
> > Regarding the question on the main benefit of SAM and if it is favorable in optimization or generalization, we argue that the primary advantage of Unified SAM lies in its flexibility, allowing it to smoothly transition between SAM and USAM by adjusting $\lambda$. The purpose of this unified framework is to provide convergence guarantees for both USAM and SAM within a single theoretical result (one for PL functions and one for general non-convex settings). Thus, the main strength of Unified SAM is its convergence guarantees and as a result, its optimization aspect. However, our experiments show that tuning $\lambda$ can also lead to improvements in generalization performance. The image classification experiments aim to demonstrate that Unified SAM performs effectively on deep learning tasks and, in some cases, even outperforms its edge cases ($\lambda=0$ or $\lambda=1$). The first half of our experiments section is devoted to verifying our theory, where we observe that, in practice, our methods work exactly as predicted by the theory. To the best of our knowledge, our work is the first to explore the practical convergence guarantees of SAM in such depth (all prior works focus mainly on providing comparisons in terms of generalization performance in settings where the theory does not necessarily hold).
> >
> > **Typos:**
> > Thank you for catching them. The paper is now updated and corrected.
> >
> >
> > **Final comment:**
> >
> > We believe the questions raised are clarification points and were easily handled in the updated version of our work (see updated PDF file). In our opinion, there is no issue related to the correctness of our results, and we hope that with our response, it becomes clear the significance of our results in terms of theory and experiments.
> >
> > We respectfully stand by our claim of correctness and significance of our results, and we politely disagree with the comment, “paper is below the bar of ICLR.” Based on the original points of the reviewer, none of the issues raised justify suggesting that the paper is below the bar of ICLR (rejection/borderline rejection).
> >
> > **If you agree that we managed to address all issues, please consider raising your mark to support our work. If you believe this is not the case, please let us know so that we have a chance to respond.**
> >
> >
> > References (not appering in the draft):
> >
> > [Kwon, 2021]: ASAM: Adaptive Sharpness-Aware Minimization for Scale-Invariant Learning of Deep Neural Networks
> >
> > [Xie, 2024]: Sampa: Sharpness-aware minimization parallelized

---

> > ### Comment · Reviewer_78w2 · 2024-11-23
> >
> > For clarification, in the updated Theorem C.4, do the authors mean **for all** $\rho\le\bar{\rho}$, $\eta\le \bar\eta$, the bound holds? or the bound just holds **for some** $\rho\le\bar{\rho}$, $\eta\le \bar\eta$. Here I am using $\bar{\rho}$ and $\bar{\eta}$ to refer to the long expression starting with $\min$ in the updated draft. The phrasing used by authors "pick $\rho\le\bar{\rho}$, $\eta\le \bar\eta$" is not precise and ambiguous here.
> >
> > My point is that it could not be the former case. $\gamma=0$ is just a extreme way to see this. If we set $\gamma$ to be $\epsilon^{10}$, the bound also breaks. Do the authors agree with this?

---

> > > ### Author Response · Authors · 2024-11-26
> > >
> > > The precise statement is “For all $\rho\leq\bar{\rho}$, $\gamma\leq\bar{\gamma}$”. We have updated that to be more precise. We kindly disagree that the bound breaks even if $\gamma$ is set to be small enough but positive (for example, $\gamma=\epsilon^{10}$ that the reviewer mentioned). In your derivation (formulas with $\Theta$), in the third bullet, you also need to take into consideration the stepsize $\rho$.
> > > Using your equation we have that the coefficient of $[f(x^0)-f^{\inf}]$ is
> > > $
> > > \frac{(1+\Theta(\gamma\rho)+\Theta(\gamma^2\rho^2)+\Theta(\gamma^2))^T}{T\gamma}\geq\frac{1+\Theta(T\gamma\rho)+\Theta(T\gamma^2\rho^2)+\Theta(T\gamma^2)}{T\gamma}=\Theta(\frac{1}{T\gamma})+\Theta(\rho)+\Theta(\gamma\rho^2)+\Theta(\gamma)
> > > $
> > > Now using the stepsize selection $\gamma=\Theta(1/\sqrt{T})$ and $\gamma=\Theta(1/\sqrt{T})$ the last expression is equal to $\Theta(\frac{1}{\sqrt{T}})+\Theta(\frac{1}{\sqrt{T}})+\Theta(\frac{1}{T\sqrt{T}})+\Theta(\frac{1}{\sqrt{T}})$ which goes to 0 as $T\to+\infty$.
> > >
> > > Essentially, the above intuitive explanation is what happens formally in the proof of Theorem 3.7 in lines 1118-1208, using the upper bound $(1+x)^T\leq\exp(Tx)$ and then forcing (by suitable choice of $\gamma$ and $\rho$) that $\exp(Tx)\leq\exp(1)$.
> > >
> > > We hope we have clarified this point and that you agree with our derivation.
> > >
> > > As we mentioned in our original rebuttal, the correctness of our theorems is not an issue, and we hope that with our response, our results' significance in terms of theory and experiments becomes clear.
> > >
> > > We respectfully stand by our claim of correctness and significance of our results, and based on the points of the reviewer, none of the issues raised justify suggesting that the paper is below the bar of ICLR (rejection/borderline rejection).
> > >
> > > **If you agree that we managed to address all issues, please consider raising your mark to support our work. If you believe this is not the case, please let us know so that we have a chance to respond.**

---

> > > > ### Comment · Reviewer_78w2 · 2024-11-27
> > > >
> > > > In equation (12), the authors write that $\frac{6\delta_0}{T\gamma} \leq \frac{\varepsilon^2}{2} \iff T \geq \frac{12\delta_0}{\gamma \varepsilon^2}$. This implies a lower bound of $\gamma$ in order for the result of Theorem C.4 to work. A too small $\gamma$, such as $\epsilon^{10}$ violates this necessary condition.

---

> > > > > ### Author Response · Authors · 2024-11-28
> > > > >
> > > > > Dear Reviewer 78w2,
> > > > >
> > > > > Thanks again for the further clarification and the detailed check of our proofs.
> > > > > There was a miscommunication in our previous response, and we appreciate the further clarification and the pointer in the exact inequality (12).
> > > > >
> > > > > We agree with the point, and we update the statement of the theorem and our proof (see pdf) to correspond to $\rho=\bar{\rho}$ and $\gamma=\bar{\gamma}$. This is a simple and straightforward update to our previous version (which included inequalities), and we agree with the reviewer that it makes it more precise and avoids any issue related to lower bounds.
> > > > >
> > > > > Thanks again for the suggestion.
> > > > >
> > > > > We hope that with the updated statement, the reviewer agrees with us on the correctness and the significance of our results and that our work in its current stage is NOT below the bar of ICLR (rejection/borderline rejection).
> > > > >
> > > > > **If you agree that we managed to address all issues, please consider raising your mark to support our work. If you believe this is not the case, please let us know so that we have a chance to respond.**

---

> > > > > > ### Comment · Reviewer_78w2 · 2024-11-28
> > > > > >
> > > > > > I thank the authors for their response. The most recent update should resolve the correctness issue and I raise my score to 5.
> > > > > >
> > > > > > But still, I didn't find the theoretical result significant. The current optimization analysis basically works in the regime where $\rho\approx 0$, where SAM works like SGD. If the goal is to get approximate stationary point/minimize the original loss, do not we directly set $\rho=0$, since $\rho$ anyway decays to $0$ with smaller $\epsilon$ or larger $T$? It is also hard to compare the optimization performance of different variant of SAM. By picking a  sufficiently small $rho$, it will eventually match the rate of SGD because it asymptotically becomes SGD.

---

> > > > > > > ### Author Response · Authors · 2024-11-29
> > > > > > >
> > > > > > > We thank the reviewer for increasing their score.
> > > > > > >
> > > > > > > We agree that having $\rho=O(1/\sqrt{T})$ is somewhat restricting however, this is a standard assumption in the SAM literature. For example:
> > > > > > >
> > > > > > > [Mi, 2022] Theorem 1: They assume *bounded gradients* **and** *bounded variance* and choose stepsizes $\rho=O(1/\sqrt{T})$ and $\gamma=O(1/\sqrt{T})$. They show that with these assumptions and parameters **SAM** converges.
> > > > > > >
> > > > > > > [Andriushchenko, 2022] Theorem 2: They assume *bounded variance* and choose stepsizes $\rho=O(1/\sqrt[4]{T})$ and $\gamma=O(1/\sqrt{T})$. They show that with these assumptions and parameters **USAM** converges.
> > > > > > >
> > > > > > > [Li, 2023] Theorem 1: They assume *bounded variance* and choose stepsizes $\rho=O(1/\sqrt{T})$ and $\gamma=O(1/\sqrt{T})$. They show that with these assumptions and parameters **SAM** converges.
> > > > > > >
> > > > > > >
> > > > > > > Our work improves the others in the following aspects:
> > > > > > >
> > > > > > > **Relaxed Assumptions:** Our main assumption, namely the Expected Residual (ER), is a more relaxed assumption and captures all the previous (bounded gradients and bounded variance) as special cases.
> > > > > > >
> > > > > > > **Unification:** Other works focus either only on SAM or USAM. Here we provide guarantees for both SAM $\lambda=1$ and USAM $\lambda=0$ as well as for any $\lambda\in(0,1)$.
> > > > > > >
> > > > > > > For the above two reasons, we believe that our theoretical contribution is of important significance in the analysis of SAM-type algorithms.
> > > > > > >
> > > > > > > We hope that with this response we clarified further the choice of $\rho$.
> > > > > > > Again, we thank the reviewer for the constructive feedback in the above discussion that helped the presentation of the paper and our results. We believe that with the above last clarification, we explain that our selection of $\rho$ makes sense. If you agree, we would appreciate increasing your score to support our work.
> > > > > > >
> > > > > > >
> > > > > > > **References:**
> > > > > > >
> > > > > > > [Mi, 2022] Peng Mi, Li Shen, Tianhe Ren, Yiyi Zhou, Xiaoshuai Sun, Rongrong Ji, and Dacheng Tao. Make sharpness-aware minimization stronger: A sparsified perturbation approach. In NeurIPS, 2022.
> > > > > > >
> > > > > > > [Andriushchenko, 2022] Maksym Andriushchenko and Nicolas Flammarion. Towards understanding sharpness-aware minimization. In ICML, 2022
> > > > > > >
> > > > > > > [Li, 2023] Bingcong Li and Georgios Giannakis. Enhancing sharpness-aware optimization through variance suppression. In NeurIPS, 2023.

---

> > > > > > > > ### Comment · Reviewer_78w2 · 2024-12-03
> > > > > > > >
> > > > > > > > I thank the authors for their citation to existing works. Still I would like to keep my current score (which has already increased from the original score in reflection of the authors' response).
> > > > > > > >
> > > > > > > > SAM with constant $\rho$, independent of number of steps and desired optimization accuracy (which is used in practice, as pointed by Reviewer hnag), is not supposed to converge to a minimizer or stationary point of the original loss, regardless of how small learning rate is and how many steps SAM has. To get a useful theory that could guide the usage of SAM in practice, we really need to analyze it in a setting/ for a goal where SGD cannot beat SAM.

---

> > > > > > > > > ### Author Response · Authors · 2024-12-04
> > > > > > > > >
> > > > > > > > > We would like to thank the reviewer for your detailed discussion, valuable feedback, and engaging back and forth throughout the review process. We appreciate your thoughtful analysis and interest in our work.
> > > > > > > > >
> > > > > > > > > We agree that having theoretical guarantees for SAM with a constant $\rho$, independent of $T$ (number of iterations) and $\epsilon$ (desired accuracy), would be ideal. However, to the best of our knowledge, no such result exists in the literature for general (non-interpolated) non-convex stochastic setting. The other works we cited (all published in major ML conferences without any issue) also depend on $T$ and rely on stronger assumptions than those presented in our paper.
> > > > > > > > >
> > > > > > > > > Furthermore, all these works, including ours, focus on providing convergence guarantees for the original loss. This is standard in theoretical convergence guarantees for SAM, and claiming that having a useful theory for SAM should only be done in a setting where SGD cannot beat SAM is undermining the whole literature on this topic.
> > > > > > > > >
> > > > > > > > > We agree that exploring guarantees in a setting where SGD can’t outperform SAM is an interesting suggestion and could be a great direction for future research.
> > > > > > > > >
> > > > > > > > > In our opinion, both a convergence analysis similar to our results and the suggestion of the reviewer are valuable, and the community should explore both of them. We hope the reviewer will agree with us on this point.
> > > > > > > > >
> > > > > > > > > Thanks again for participating in a back-and-forth discussion with us.

---

### Official Review · Reviewer_Mo81 · 2024-11-04

**Soundness:** 3
**Presentation:** 4
**Contribution:** 3
**Rating:** 6
**Confidence:** 4

**Summary:**

This paper extends and combines previous analyses on the convergence of SAM and unnormalized SAM (U-SAM) by considering a generalized update rule called Unified SAM. Under the Expected Residual Condition, they prove convergence for Unified SAM for loss functions satisfying PL conditions and generalized non-convex loss functions. The convergence bound holds for a wide range of sampling strategies. Intriguingly, they show that importance sampling can minimize this convergence bound. Empirically, they show that Unified SAM matches and sometimes outperforms the original SAM.

**Strengths:**

[Quality] This paper is well-written and contains solid theoretical and empirical results.
[Clarity] This paper carefully discusses previous convergence bounds on SAM and U-SAM and shows clearly how the current convergence results generalize previous ones.

**Weaknesses:**

The connections and distinctions between this work and previous analyses of SGD require clearer articulation. The paper’s central assumption, the Expected Residual Condition, is adopted from [1] and is also employed in [2,3]. In the limiting case where $\rho = 0$, the results presented here converge to those found in the SGD analyses in [2,3]. While the authors make this reduction explicit, the paper does not address how the implications of the current results and proof techniques differ from those of earlier studies.
For instance, the authors suggest that this work "provides a theoretical justification for applying importance sampling in SAM." However, this argument is based on the quantity $\max_i L_i/p_i$ within the bound, which also appears in previous analyses, such as [1]. Thus, this justification extends to SGD as well and is not specifically related to SAM, a point that should be conveyed more directly.
The connection between the current theoretical and empirical results could be further strengthened. The computer vision experiments in this paper demonstrate that Unified SAM can achieve improved validation performance. However, since the primary focus of this paper is on the convergence properties of these methods, training loss would serve as a more relevant metric for linking the theory with empirical findings. This metric, however, is not included in the paper or its appendix.
The presentation can be improved.
The current paper contains typos that may obscure understanding. For example, the $g(x)$ in equation (ER) in Assumption 3.1 should be the norm of $g(x)$. Further, given the complexity of the current theoretical bounds, an intuitive interpretation of the current bounds can improve the paper.

[1] SGD: General Analysis and Improved Rates. arxiv.org/abs/1901.09401
[2] SGD for Structured Nonconvex Functions: Learning Rates, Minibatching and Interpolation arxiv.org/abs/2006.10311
[3] Better Theory for SGD in the Nonconvex World. arxiv.org/abs/2002.03329

**Questions:**

Questions
[Relationship of convergence speed and $\lambda$] In the logistic regression experiments, the convergence is better with larger $\lambda$, which is coherent with the theory as here $C = 0$. Is there a case that the bound will be minimized at a non-zero $\lambda$ when $C \neq 0$ and in general, is there a setting where Unified SAM’s convergence speed will strictly improve over U-SAM?
[Empirical Studies] In the case of computer vision experiments, is the training loss of Unified SAM or USAM lower than SAM? As the best practice so far seems to be using SAM in the later phase of training, how to disentangle the sharpness reduction benefit of SAM over U-SAM [1,2]?


[1] The Crucial Role of Normalization in Sharpness-Aware Minimization, arxiv.org/abs/2305.15287
[2] How Does Sharpness-Aware Minimization Minimize Sharpness? arxiv.org/abs/2211.05729

---

> ### Author Response · Authors · 2024-11-20
> **Authors' response to Reviewer Mo81**
>
> We would like to thank the reviewer for their time and the positive evaluation. We appreciate the comments on the strengths of our work and the characterization of having solid theoretical and empirical results.
>
> Below, we address the concerns raised by the reviewer.
>
> **This work and previous analyses of SGD:**
>
> As we mentioned in our paper (lines 156-158) as corollaries of the main theorems on the analysis of SAM for the two classes of problems we focus on (PL and non-convex problems), we obtain the state-of-the-art convergence guarantees for SGD (for ρ = 0), showing the tightness of our analysis.
>
> The two methods (SGD and SAM) are conceptually different. As we explained in our paper, SAM is proposed as a method for direct sharp minima during the training process, while SGD does not necessarily possess such property. This is possible as we allow having positive \rho in the update of SAM (that itself can be interpreted as a solver of the min-max problem - see line 056).
>
> The proofs for the convergence guarantees of the two methods are substantially different from each other. For example, one important difference is that for the analysis of SAM, one needs to handle gradient norms and inner products that do not appear in the analysis of SGD (see lemmas A6 and A7 in the appendix for precise statements).
>
> **On Importance sampling:**
>
> The quantity $\max_i L_i / np_i$ naturally arises in almost all stochastic methods for solving smooth optimization problems. See for example [Gower, 2019; Khaled, 2020] for SGD and [Choudhury, 2024] for variational inequalities. So, it’s not surprising (it is actually expected) that it appears in the analysis of SAM as well. In most papers focusing on the analysis of stochastic methods, the quantity $\max_i L_i / np_i$ as a lower bound on the number of iterations required to achieve specific accuracy. As we explained in Section 3.4 this is the reason that the probability $p_i = L_i / \sum_{j=1}^n L_j$ should be used instead of the more classical uniform sampling.
>
> **On Loss Plots in experiments:**
>
> The reviewer mentioned, “since the primary focus of this paper is on the convergence properties of these methods, training loss would serve as a more relevant metric for linking the theory with empirical findings.”
>
> For this, let us note that the first half of our experiments, in Section 4.1, Figs 1, 2, 3, are all training loss plots (exactly related to what the reviewer requested). In these plots, we verify exactly the theoretical findings of our work.
> Other works on the analysis of SAM-type methods have often focused on theoretical analyses and empirical results without the use of explicit loss plots; see [Li & Giannakis, 2023; Zhuang et al, 2022; Mi et al, 2022] (only present tables with generalization performance). The focus of our work is providing formal proofs and empirical benchmarks, which, in our opinion, offer a clearer assessment of SAM's performance across theoretical and practical scenarios.
> We agree with the reviewer, and this is precisely the reason why, in our submission, we included loss plots as well.
>
> **Typo:**
> Thank you for catching the typo. It is fixed.
>
> **Theoretical Benefit of Unified SAM’s over U-SAM:**
>
> This is indeed an interesting question. In our work, we focus primarily on exploring the convergence of SAM, and U-SAM and their potentially interesting in-between variants ($\lambda \in (0,1)$ under relaxed assumptions and meaningful step-size selections. However, based on our current theoretical results, providing a closed-form expression of the best $\lambda$ is a challenge by itself (a different optimization problem). In this work, our primary goal is to show that more variants beyond the two most popular SAM and USAM can be analyzed and have beneficial practical performance (see experiments on Unified SAM). A possible future direction will be to describe when Unified SAM is theoretically advantageous against USAM or SAM (with strong theoretical guarantees).
>
> **Empirical Studies:**
> In the computer vision experiments, there are cases where the training loss of Unified SAM is lower than SAM, and vice-versa. We can easily include such plots in the camera-ready version (experiments have already been run).
> Regarding the final question, “How to disentangle the sharpness reduction benefit,” we are unsure what the reviewer meant. Can you please clarify?
>
> Thanks again for the review and the positive evaluation of our work.
> Reading your comments, we believe that all pointed weaknesses are simple clarifications.
> **If you agree that we managed to address all issues, please consider raising your mark to support our work. If you believe this is not the case, please let us know so that we have a chance to respond.**

---

> > ### Comment · Reviewer_Mo81 · 2024-11-28
> > **reply**
> >
> > I have read the response and thank the authors for the clarification.
> > I believe it is important to concretely quantify how the convergence speed is impacted by \lambda in Unified SAM empirically in the computer vision experiments. This could then reveal how exactly Unified SAM improves over SAM / USAM, whether it is by improving optimization or by improving generalization. As stated in the review, I think this discussion can improve the paper and hope to see it in future versions.
> > I will keep my positive rating for this paper.

---

### Official Review · Reviewer_mscb · 2024-11-04

**Soundness:** 3
**Presentation:** 4
**Contribution:** 3
**Rating:** 8
**Confidence:** 2

**Summary:**

The authors provide a Unified framework (Unified SAM) as a convex combination of SAM ascend and USAM ascend. They provide convergence guarantees for Unified SAM via PL condition. In special cases, the bounds reduce to that of SGD, which is known. The sampling they consider in their method is arbitrary, not restricted, and includes importance sampling. The paper concludes with experiments.

**Strengths:**

- interesting and well-motivated problem
 - very well written
 - clean theoretical contributions and supporting experiments

**Weaknesses:**

- the paper would benefit from an informal statement of results (in math) in terms of convergence rates at the begining

**Questions:**

This is an interesting paper. I have a question: how does the rate you prove depend on the parameter $\lambda$? I understand that you derive the results in the paper, but can you explain in words how the convergence rate is changed when $\lambda$ varies? Another question is, are you making the previous bounds tighter, or do you present a proof that works for new regimes?

---

> ### Author Response · Authors · 2024-11-20
> **Authors' response to Reviewer mscb**
>
> We thank the reviewer for the review and positive evaluation.
>
> Below, we address the questions raised by the reviewer.
>
> Thanks for the suggestion. Let us highlight that we do have a table (Table 1) that has informal statements of our theorems and other related works.
>
> **How the rates change as $\lambda$ varies:**
>
> The choice of $\lambda$ indeed affects the performance of the method. To understand the connections of $\lambda$ and the two step-sizes of the algorithm $\rho$ and $\gamma$, let us focus on the PL problems, one class of problems we focus on in this work (similar statements can be obtained for nonconvex as well - but the connection are arguably more complicate to present).
>
> We have the following dependence between the step-sizes $\rho$ and $\gamma$ of the proposed method, and the parameter $\lambda$. In particular, we have that $\rho=O(1/(1-\lambda)^2)$ and $\gamma=O((1-\lambda)^2)$, so when $\lambda$ increases from 0 to 1, $\rho$ increases while $\gamma$ decreases. As a result, with the increase of $\lambda$, the convergence of the method is slower. However,  as we highlighted in Thm 3.2, any $\lambda \in [0,1]$, we always have linear convergence.
>
>
> **About the bounds and proofs:**
>
> Our work relaxes conditions used in previous analyses and, at the same time, provides new and improved convergence rates for both USAM and SAM. Furthermore, we extend the flexibility of sampling selection in SAM-type methods (via the unified SAM approach) as we provide an analysis under the arbitrary sampling framework that includes important sampling as a special case (sampling strategy that improves the theoretical complexity of our theorems over the more classical uniform sampling).

---

> > ### Comment · Reviewer_mscb · 2024-11-25
> >
> > Thanks for your response to my comments! If space permits, please also consider revising the paper to add more discussion on them. I decided to keep my score unchanged.

---

### Author Response · Authors · 2024-11-20
**General response to all reviewers**

We thank the reviewers for their feedback and time.

In particular, we appreciate that the reviewers acknowledged the following strengths of our work:
* Reviewer **mscb** finds that we tackle an interesting and well-motivated problem, and we have clean theoretical contributions and supporting experiments.
* Reviewer **Mo81** appreciates the quality and clarity of our paper and that it contains solid theoretical and empirical results.
* Reviewer **78w2** recognizes that our theoretical results for convergence of SAM under the Expected Residual (ER) Condition are new.
* Reviewer **hnag** acknowledges that our paper is well-written and easy to follow.

With our rebuttal, we address all raised issues. **Here we highlight again that with our work:**

* We propose the **Unified SAM**, an update rule that is a convex combination of SAM and USAM. The new formulation captures both USAM and SAM as special cases, but more importantly, it opens up a wide range of possible update rules.
* We provide **convergence guarantees** for Unified SAM, for smooth functions satisfying the PL condition, and for general non-convex functions.
* We extend our convergence guarantees of Unified SAM to under **arbitrary sampling**. This allows us to cover a wide range of samplings for USAM and SAM that were never considered previously in the literature.
* All the provided convergence guarantees are **tight** in the following sense: If $\rho=0$ Unified SAM reduces to SGD and our theorems recover as a special case the best-known convergence rates of non-convex SGD.
* Finally, we have extensive **numerical evaluations** where we validate our theoretical results and evaluate the proposed methods in training DNNs.

**We hope that you will engage with us in a back-and-forth discussion, and we will be most happy to answer any remaining questions.**

In the updated version of our submitted PDF, we fixed all typos mentioned and corrected the statement of one of Theorem 3.7 and its proof to reflect the comment of reviewer 78w2.

---

### Meta-Review · Area_Chair_bd9w · 2024-12-21

**Metareview:**

This paper focuses on addressing some of the open problems related to SAM's convergence properties, including the role of normalization, the impact of noise assumptions, and the effect of different sampling strategies. The authors introduce a unified framework for SAM that encompasses both normalized and unnormalized SAM, and establish convergence guarantees for this framework.The analysis accommodates arbitrary sampling strategies, enabling the study of previously unexplored SAM variants. Experiments validate the theoretical findings and demonstrate the effectiveness of the unified SAM framework in training deep neural networks for image classification. Reviewers appreciate the paper's theoretical contributions and experimental support, recognizing its value in enhancing our understanding of SAM's behavior in non-convex optimization.

The reviewers also provided suggestions for improvement and raised some concerns about the paper. Reviewer mscb suggested that the authors include an informal mathematical statement of the results at the beginning to provide better intuition about the interconnectedness of the results developed throughout the paper. The reviewer also asked questions about the bounds, proofs, and how the rates change as lambda varies. The authors responded to these questions and highlighted that Table 1 already contains informal statements of their theorems and those from related works. Reviewer Mo81 had concerns about the connections and distinctions between this work and previous analyses of SGD. The authors responded by explaining the differences in their proof technique. The reviewer also believed it is important to empirically quantify how lambda in Unified SAM impacts convergence speed. Reviewer 78w2 carefully checked the proofs of the theoretical results and found a couple problematic arguments. This led to several rounds of feedback and responses between the reviewer and the authors until the issues were resolved through corrections. As a result, the reviewer increased their initial rating of the paper. However, the reviewer believes the theory could be more useful if it covered a setting where SAM outperforms SGD. Reviewer hnag questioned the assumption about the perturbation radius, stating that the authors' proof assumes this radius diminishes to zero, whereas in practical training it does not. The reviewer also argued that the lack of comparison with Nam et al. [2023] makes assessing the contributions of this work difficult due to similarities between the two. The authors adequately responsed to these questions. In particular, regarding the first concern about the diminishing radius, they clarified that in all their theoretical results (except Theorem 3.5, which focuses on PL functions), the radius parameter is constant, not decreasing.

Overall, the paper is borderline accept with ratings ranging from 5 to 8. However, I believe the strengths outweigh the weaknesses, and that the unified SAM framework and the presented results can make meaningful contributions to our understanding of SAM's convergence properties. Therefore, I recommend accept.

**Additional Comments On Reviewer Discussion:**

Reviewer mscb suggested that the authors include an informal mathematical statement of the results at the beginning to provide better intuition about the interconnectedness of the results developed throughout the paper. The reviewer also asked questions about the bounds, proofs, and how the rates change as lambda varies. The authors responded to these questions and highlighted that Table 1 already contains informal statements of their theorems and those from related works. Reviewer Mo81 had concerns about the connections and distinctions between this work and previous analyses of SGD. The authors responded by explaining the differences in their proof technique. The reviewer also believed it is important to empirically quantify how lambda in Unified SAM impacts convergence speed. Reviewer 78w2 carefully checked the proofs of the theoretical results and found a couple problematic arguments. This led to several rounds of feedback and responses between the reviewer and the authors until the issues were resolved through corrections. As a result, the reviewer increased their initial rating of the paper. However, the reviewer believes the theory could be more useful if it covered a setting where SAM outperforms SGD. Reviewer hnag questioned the assumption about the perturbation radius, stating that the authors' proof assumes this radius diminishes to zero, whereas in practical training it does not. The reviewer also argued that the lack of comparison with Nam et al. [2023] makes assessing the contributions of this work difficult due to similarities between the two. The authors adequately responsed to these questions. In particular, regarding the first concern about the diminishing radius, they clarified that in all their theoretical results (except Theorem 3.5, which focuses on PL functions), the radius parameter is constant, not decreasing.

---

### Decision · Program_Chairs · 2025-01-22

Accept (Poster)